# UniMedVL: Unifying Medical Multimodal Understanding and Generation through Observation-Knowledge-Analysis

## Abstract

Clinical diagnosis demands models that can process multimodal medical inputs (images, patient histories, lab results) and generate diverse outputs including both textual reports and visual content (annotations, segmentation masks, and images). Despite this need, existing medical AI systems disrupt this unified process: medical image understanding models interpret images but cannot generate visual outputs, while medical image generation models synthesize images but cannot provide textual explanations. This leads to gaps in data representation, feature integration, and task-level multimodal capabilities. To this end, we propose a multi-level framework that mirrors clinical diagnosis through the Observation-Knowledge-Analysis (OKA) paradigm. Specifically, at the observation level, we construct **UniMed-5M**, a dataset comprising over 5.6M samples that reformat diverse unimodal data into multimodal pairs for foundational observation. At the knowledge level, we propose **Progressive Curriculum Learning** that systematically introduce medical multimodal knowledge. At the analysis level, we introduce **UniMedVL**, the first medical unified multimodal model for the simultaneous analysis of image understanding and generation tasks within a single architecture. UniMedVL achieves superior performance on five medical image understanding benchmarks, while matching specialized models in generation quality across eight medical imaging modalities. Crucially, our unified architecture enables bidirectional knowledge sharing generation tasks enhance visual understanding features, demonstrating that integrating traditionally separate capabilities within a single medical framework unlocks improvements across diverse clinical scenarios. Code is available at Link.

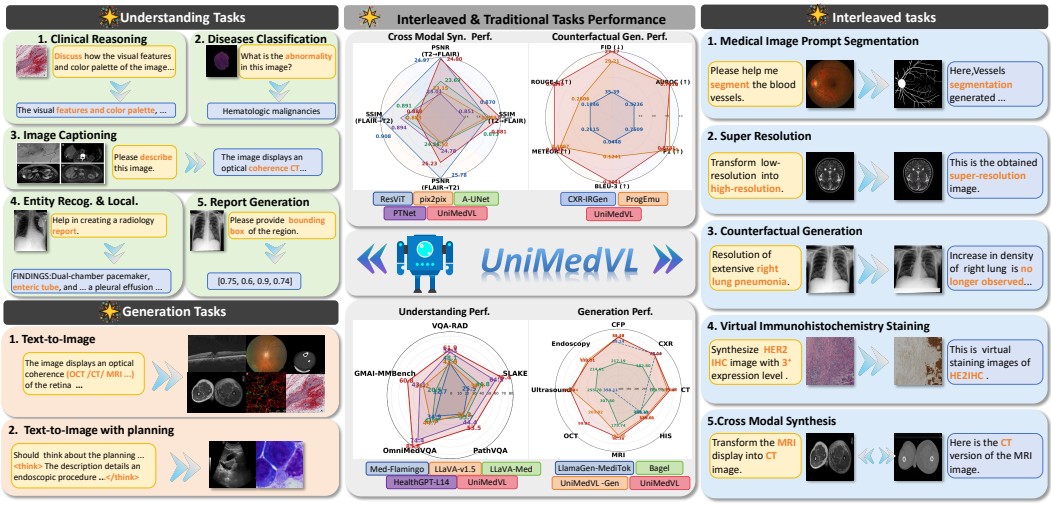

Figure 1: **Overview of UniMedVL unified framework.** Capabilities across medical image understanding and generation tasks and performance comparisons.

## 1 INTRODUCTION

Clinical diagnosis fundamentally follows a structured multi-level reasoning pipeline that is inherently multimodal in both inputs and outputs. Physicians systematically **observe** multimodal raw data (imaging patterns, patient histories, symptom descriptions (Huang et al., 2020)), integrate this with clinical **knowledge** (medical literature, domain expertise, cross-modal associations (Khader et al., 2023)), and **analyse** to produce diverse diagnostic outputs–textual reports explaining findings, visual annotations localizing abnormalities, and comparative imagery for treatment planning (Nguyen et al., 2023; Tanida et al., 2023).

Consider a radiologist examining suspected lung pathology: they process chest X-rays (visual), prior CT scans (cross-modal comparison), and patient history (textual) to generate multiple complementary outputs: detailed reports describing findings, visual annotations highlighting specific regions, and comparative visualizations for surgical planning. This exemplifies how clinical diagnosis requires unified processing of multimodal inputs to generate diverse multimodal outputs, where neither textual reports alone (lacking spatial localization) nor visual annotations alone (lacking reasoning context) suffice. While multimodal fusion has demonstrated substantial improvements in clinical decision-making (Benani et al., 2025; Soenksen et al., 2022), current medical AI remains fragmented, with state-of-the-art models achieving less than 60% accuracy compared to over 90% for human experts on diagnostic challenges (Kaczmarczyk et al., 2024). This fragmentation manifests at three critical levels: (i) **Data**: Medical datasets remain predominantly single-modal despite clear evidence that multimodal integration substantially improves diagnostic accuracy (Warner et al., 2024; Huang et al., 2023). (ii) **Features**: Current approaches lack systematic progressive training strategies that can effectively capture deep cross-modal relationships—most methods simply concatenate features rather than progressively building from basic pattern recognition to sophisticated multimodal reasoning (Haq et al., 2025). (iii) **Tasks**: While general-domain models have made progress in unified architectures, the medical domain still lacks truly unified models. For instance, although HealthGPT demonstrates both understanding and generation capabilities for medical tasks, it requires reloading different model checkpoints to switch between task types—a limitation that prevents seamless multi-task operation in clinical workflows (Lin et al., 2025).

To bridge this gap, we propose a workflow-guided framework that mirrors how physicians actually process medical information through the Observation-Knowledge-Analysis (OKA) paradigm: Observation Level (Data): We construct UniMed-5M, a 5-million sample dataset that, unlike existing single-modal datasets, reformats medical data of various tasks into true multimodal input-output compatible pairs. Knowledge Integration Level (Features): We design Progressive Curriculum Learning that goes beyond naive concatenation. Through three carefully designed stages (alignment for medical data, fusion, and synthesis), our approach materialises models to discover cross-modal patterns better. Analysis Level (Tasks): We introduce UniMedVL, the first unified medical model capable of both understanding and generation within a single architecture at the same time. Our experiments validate two key insights: (1)Medical Representation learning requires framework, allowing the shared representations in medical domains necessitate the OKA framework with sufficient data quality and quantity, for unified medical multimodal learning; (2) Rapid adaptation is achievable, unified model architectures demonstrate the feasibility of quickly adapting to new medical tasks and datasets for scalable multimodal medical AI intelligence. Therefore, our contributions are as follows:

- **Observation (Data-level):** We construct **UniMed-5M**, a large-scale dataset containing over 5.6M multimodal medical examples that reformat diverse unimodal datasets into uniform multimodal input-output pairs, and serve as the initial building blocks for unifying diverse medical tasks.

- **Knowledge integration (Feature-level):** We devise **Progressive Curriculum Learning**, a three-stage training paradigm that systematically builds medical multimodal capabilities: foundation training for basic pattern recognition, instruction tuning for cross-modal fusion, and unified multi-modal training for advanced synthesis.

- **Analysis (Task-level):** We introduce **UniMedVL**, a novel unified medical foundation model that provides multimodal capabilities–including understanding multimodal inputs and generating textual reports, image translation, segmentation masks, and synthetic medical images–within a single architecture without needing offline checkpoints once loaded.

## 2 Related Work

### 2.1 Medical Multimodal Large Language Models

Early medical MLLMs took an adapter-style approach, plugging medical vision encoders into general LLMs through lightweight projection or LoRA. Thawakar et al. (2024) aligned MedCLIP with Vicuna via a linear projector in XrayGPT. Li et al. (2023) bootstrapped instruction data from PubMed figures using GPT-4 in LLaVA-Med. These systems proved effective for VQA and reporting but kept fusion shallow and provided no unified route to image synthesis or editing. A second paradigm emphasizes data engineering. Chen et al. (2024b) leveraged GPT-4V to reformat noisy PubMed image–text pairs into the 1.3M-sample PubMedVision corpus in HuatuoGPT-Vision. While this approach mitigates data scarcity and noise, it remains primarily comprehension-oriented; unified, high-fidelity generation is still outside the model proper. Zhang et al. (2023a), which extends beyond radiology to molecules and proteins in a unified seq2seq architecture with BioMedGPT, and Singhal et al. (2025), which reaches expert-level exam performance via chain-of-thought and clinician-aligned instruction with Med-PaLM 2, strengthen biomedical reasoning but do not deliver a unified medical pipeline that natively spans image-level generation and text reasoning. More recently, Lin et al. (2025) emerged with HealthGPT as the first medical MLLM explicitly targeting unified multi-modal input and output. It introduces MOE LoRA to reduce task interference and to cover tasks. However, its unification relies on multiple task-specific models at inference time. As a result, different capabilities are not consolidated into a single, end-to-end model that uniformly expresses all tasks simultaneously.

### 2.2 Unified Multimodal Understanding and Generation Models

Outside the medical domain, unified multimodal research has developed along several paradigms. Autoregressive models (Team, 2024a; Wang et al., 2024; Lu et al.; 2024) treat images as discrete tokens within decoder-only Transformers, achieving architectural unity but limiting high-resolution synthesis due to long sequences and discrete reconstruction. Dual-encoder designs (Wu et al., 2025c; Ma et al., 2025d; Xu et al., 2025b) address the granularity conflict between semantic understanding and pixel-level generation through separate visual pathways, improving task-specific performance at increased inference cost. Hybrid objectives combine different generative paradigms: Zhou et al. (2024) jointly optimize text cross-entropy and image diffusion losses in Transfusion, Xie et al. (2024) embed discrete diffusion via Omni-Attention for faster synthesis in SHOW-O, while modular approaches (Wu et al., 2025e; **?**; 2024a) bridge frozen MLLMs with diffusion models through learnable connectors. These solutions achieve cost-effectiveness but sacrifice end-to-end differentiability. Representation innovations target the semantics-fidelity gap through various strategies: multi-codebook quantization (Ma et al., 2025c), contrastive-aligned tokenization (Wu et al., 2024b), unified CLIP semantic spaces (Chen et al., 2025), and masked autoregressive encoders (Jiang et al., 2024). Advanced autoregressive methods (Liao et al., 2025; Zhang et al., 2025; Zhuang et al., 2025) enable high-fidelity interleaved generation through deep fusion, prefilled tokens, and reinforcement learning from human feedback. Despite these advances, balancing semantic understanding with pixel-level reconstruction remains challenging, particularly for fine-grained medical localization and diagnostic-quality image synthesis required in clinical applications.

## 3 Methodology

Our workflow-guided multi-level framework systematically implements the clinical Observation-Knowledge-Analysis (OKA) paradigm through three corresponding stages: data-level observation for comprehensive multimodal dataset construction, feature-level knowledge integration through principled curriculum learning, and task-level analysis via unified model architecture. Each stage addresses specific computational challenges while maintaining clinical workflow coherence.

### 3.1 Observation Level: UniMed-5M Dataset Construction

At the observation level, comprehensive multimodal datasets are constructed to enable systematic processing of diverse medical inputs that mirror clinical practice. The dataset construction follows clinical workflow patterns where multiple data modalities are observed and initially processed before knowledge integration. The overall dataset curation pipeline is shown in Fig. 2.

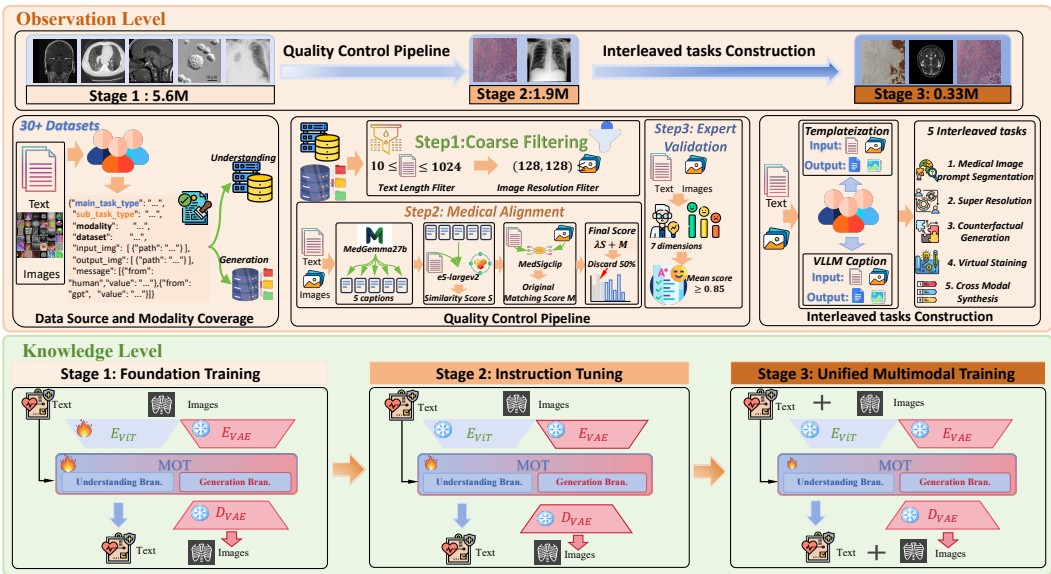

Figure 2: Overview of the proposed **Observation–Knowledge** framework. **Observation**: Covers data sources and modality coverage, quality control pipeline, and interleaved task construction for building training data across different model stages. **Knowledge**: Refers to the progressive curriculum training paradigm, consisting of three stages that gradually equip the model with generalized capabilities on interleaved tasks.

**Data Source and Modality Coverage.** A comprehensive medical dataset comprising 5.6M samples is assembled from diverse public repositories including PMC-OA, Quilt-1M, PubMedVision, GMAI-VL datasets, CheXpertPlus, PMC-VQA, Medical-Diff-VQA, SLAKE, PathVQA, and other specialized medical datasets through systematic data synthesis and augmentation methodologies detailed in Appendix A.2. The collection encompasses nine primary imaging modalities: chest X-rays (CXR), histopathology images (HIS), CT scans, MRI sequences, retinal fundus photography (CFP), optical coherence tomography (OCT), endoscopy, ultrasound, and fluorescence microscopy (FM). The dataset encompasses diverse medical AI task categories spanning understanding, generation, and multimodal input-output capabilities.

**Quality Control Pipeline.** We adopt a three-step pipeline that progressively increases fidelity while controlling cost:

- **Coarse Filtering.** Images are preprocessed through modality-specific normalization and resolution filtering ($\geq$ 128×128 pixels). Text undergoes specialized tokenization that preserves medical terminology, followed by length filtering (16–1024 characters).

- **Medical Alignment.** Because medical captions often emphasize specific pathological findings rather than exhaustive descriptions, we implement a dedicated verification pipeline. MedGemma-27b (Sellergren et al., 2025) generates five diverse captions per image; semantic similarity is computed with E5-large-v2 embeddings (Wang et al., 2022); and medical-specific alignment is assessed using MedSigCLIP (Sellergren et al., 2025). We then compute a combined alignment score $\text{score}_{\text{final}} = \lambda \cdot \text{similarity}_{\text{E5}} + \text{score}_{\text{MedSigCLIP}}$ with $\lambda = 0.5$, retaining the top 50% of pairs as high-quality training data.

- **Expert Validation.** Medical experts draw a 5% stratified subset for comprehensive auditing strictly along the seven dimensions: modality–image match, factual accuracy, information completeness, position/quantity accuracy, professionalism, planning coherence, and clinical reasoning quality. This stage serves as quality assurance rather than additional filtering: on the audited subset, expert ratings across all dimensions are observed to be $\geq$0.85 (with high inter-rater agreement), substantiating the dataset's clinical reliability and mitigating potential LLM hallucinations.

**Interleaved tasks Construction.** This component encompasses five interleaved tasks: medical image prompt segmentation, super-resolution, counterfactual generation, virtual staining, and cross-modal

synthesis. We adopt two complementary construction strategies: Templateization and VLLM Caption. In Templateization, inputs and outputs are standardized into structured image–text pairs, where textual prompts explicitly guide the model beyond the provided image and outputs follow a templated format. In contrast, VLLM Captioning emphasizes generating semantically rich textual descriptions that interpret the corresponding images in medical contexts, including symptom analysis and clinical insights.

## 3.2 Knowledge Level: Progressive Curriculum Learning

At the knowledge integration level, deep cross-modal knowledge fusion is achieved through a principled curriculum learning paradigm that progressively builds from basic medical pattern recognition to sophisticated multimodal reasoning capabilities.

**Progressive Curriculum Training Paradigm:**

- **Stage 1: Foundation Training.** Foundational medical domain awareness is established through unsupervised exposure to comprehensive medical datasets. The foundation training stage prioritizes broad pattern recognition over task-specific performance, enabling robust medical concept acquisition through text-image paired learning and next-token prediction across diverse medical sources. Furthermore, the training emphasizes learning general medical visual-language alignments without task-specific constraints and overly curated datasets.

- **Stage 2: Instruction Tuning.** Medical expertise is systematically developed through fine-tuning on curated high-quality instruction data. The instruction-formatted medical tasks follow the format $(q, x_v, k) \rightarrow (a_t, a_v)$ where query $q$, visual input $x_v$, and knowledge context $k$ generate textual $a_t$ and visual $a_v$ responses. We implement differentiated enhancement strategies for distinct task types: For medical understanding tasks such as VQA, we augment standard responses with existing Distilled Chain of Thought (DCOT) data that explicitly articulate the reasoning pathway from visual observation to medical conclusions. For generation tasks, we employ Caption Augmented Generation (CAG) pipeline to enhance caption quality, incorporating structured planning steps that guide the visual synthesis process. The details are provided in Appendix A.5.

- **Stage 3: Unified Multimodal Training.** Multimodal capabilities of generation and understanding are developed through sophisticated tasks requiring integrated visual-textual reasoning. This stage focuses on complex interleaved tasks that combine understanding and generation requirements within unified sequences. The training strategy maintains semantic stability from previous stages while enabling advanced synthesis capabilities through selective parameter optimization, preparing the model for comprehensive clinical analysis workflows.

## 3.3 Analysis Level: UniMedVL Unified Architecture

At the analysis level, comprehensive multimodal medical outputs are generated through a unified architecture that naturally simulates clinical analytical processes while maintaining cross-modal consistency. The UniMedVL architecture integrates the progressive curriculum learning paradigm into a cohesive system capable of both understanding and generation within a single model.

**Task Organization.** Model training is systematically organised into three primary tasks that reflect fundamental capabilities required for unified medical multimodal systems: (i) Understanding tasks encompassing medical image comprehension, VQA, diagnostic reasoning, image captioning, and clinical report generation; (ii) Generation tasks focusing on text-to-image synthesis with conditional medical image generation and planning-guided approaches; and (iii) Interleaved tasks combining visual-textual inputs and outputs requiring seamless multimodal integration. These interleaved tasks include sophisticated capabilities such as virtual immunohistochemistry staining , cross-modal synthesis (e.g., CT to MRI synthesis), counterfactual generation for treatment planning and development forecasting.

**Model Architecture Overview.** Following Deng et al. (2025), we adopt a unified architecture with dual visual encoders and mixture-of-transformer-experts (MoT). The understanding-oriented encoder $E_{ViT}$ extracts semantic tokens $z_{ViT} = E_{ViT}(x_v)$ for multimodal comprehension tasks, while the generation-oriented encoder $E_{VAE}$ produces latent representations $z_{VAE} = E_{VAE}(x_v)$ for visual synthesis tasks. The MoT module contains specialized transformer experts: an understanding expert

processes interleaved sequences of text and ViT tokens $[x_{text}, z_{ViT}]$ for vision-language understanding, while a generation expert handles sequences containing text and VAE tokens $[x_{text}, z_{VAE}]$ for image generation. Projection layers $f_{ViT}$ and $f_{VAE}$ bridge the visual encoders with the transformer experts, mapping encoded features to the shared hidden dimension. For generation outputs, the decoder $D_{VAE}$ reconstructs visual content from the latent representations back to pixel space. A generalized causal attention mechanism enables both experts to operate on the same token sequence through shared self-attention operations for different types of tasks, integrating understanding and generation paradigms into a unified framework.

**Training Objectives.** The model is trained with a unified loss function combining understanding task loss and generation task loss. For understanding tasks, we employ next-token prediction:

$$\mathcal{L}_{\text{NTP}} = -\sum_{i=1}^{n} \log p(t_{i+1}|t_{\leq i}, z_{VIT}; \theta), \tag{1}$$

where $t_i$ denotes the $i$-th text token and $\theta$ represents model parameters. For visual generation, rectified flow matching is applied on VAE latents:

$$\mathcal{L}_{\text{flow}} = \mathbb{E}_{t,\epsilon}[\|v_\theta(\tilde{z}_{VAE}, t, c) - v\|^2], \tag{2}$$

where $v_\theta$ is the velocity prediction network, $\tilde{z}_{VAE}$ represents noisy latents processed by $D_{VAE}$, $t$ is the time step, $c$ denotes conditioning, and $v$ is the target velocity field. The overall training loss is as follows:

$$\mathcal{L} = \mathcal{L}_{\text{NTP}}(z_{VIT}) + \alpha \cdot \mathcal{L}_{\text{flow}}(z_{VAE}), \tag{3}$$

where the coefficient $\alpha$ balances the contribution of generation tasks. We set $\alpha = 4$ empirically.

## 4 EXPERIMENTS

### 4.1 BENCHMARKS AND BASELINES

**Evaluation Benchmarks.** We evaluate UniMedVL across medical visual understanding and generation benchmarks. For **image understanding tasks**, we employ VQA-RAD (Lau et al., 2018), SLAKE (Liu et al., 2021), PathVQA (He et al., 2020), OmniMedVQA (Hu et al., 2024), and GMAI-MMBench (Ye et al., 2024), which cover diverse medical scenarios. For **image generation tasks**, we split the image–caption pairs in the proposed dataset into 80% for training and 20% for testing. We use the test set to evaluate UniMedVL's text-to-image generation performance. For **interleaved tasks**, we utilize the BCI dataset (Liu et al., 2022b) for the virtual immunohistochemistry staining task. The IXI dataset (**?**) is leveraged to evaluate the super-resolution task, and the BraTS 2023 dataset (Adewole et al., 2023) is used for evaluating the cross-modal synthesis task. We use the ICG-CXR dataset (Ma et al., 2025b) to evaluate the counterfactual generation task.

**Baseline Methods.** These include two categories of methods: **specialized models** and **unified multimodal models**. For specialized models, we include understanding-only models such as Med-Flamingo (Moor et al., 2023), LLaVA-Med (Li et al., 2023), HuatuoGPT-Vision (Chen et al., 2024b), RadFM (Wu et al., 2025b), GMAI-VL (Li et al., 2024), LLaVA-v1.5 (Liu et al., 2024), and InternVL2 (Team, 2024b). We also compare with image translation models including CycleGAN (Zhu et al., 2017), pix2pix (Isola et al., 2017), pix2pixHD (Wang et al., 2018), pyramid pix2pix (Liu et al., 2022b), SRCNN (Dong et al., 2015), VDSR (Kim et al., 2016), SwinIR (Liang et al., 2021), Restormer (Zamir et al., 2022), AMIR (Yang et al., 2024), ResViT (Dalmaz et al., 2022), and TransUNet (Chen et al., 2021). Additionally, to determine the model performance of medical imaging generation capability, we finetuned LlamaGen-MediTok (Ma et al., 2025a). For unified multimodal models, we include general frameworks like Janus (Wu et al., 2025d) and Bagel (Deng et al., 2025), as well as medical unified models such as HealthGPT (Lin et al., 2025).

**Evaluation Metrics.** We employ task-specific metrics aligned with clinical relevance. For **image understanding tasks**, we utilize accuracy as the evaluation metric. For **image generation tasks**, we employ generation FID (gFID) and BioMedCLIP score to evaluate the quality of synthesized images. For **interleaved tasks**, we leverage PSNR and SSIM as evaluation metrics for virtual immunohistochemistry staining, super-resolution, and cross-modal synthesis tasks. For counterfactual generation, we follow the experimental setup of ProgEmu (Ma et al., 2025b), using gFID, AUC-ROC, and F1 to evaluate the quality of synthesized images, and BLEU-3, METEOR, and ROUGE-L to assess the quality of the explanatory text.

Table 1: **Comparison of UniMedVL with other LVLMs and unified multi-modal models on medical visual understanding tasks. Bold** and underlined text indicate the best performance and second-best performance, respectively.

| Model | Params | Medical | VQA-RAD | SLAKE | PathVQA | OmniMedVQA | GMAI-MMBench |
|---|---|---|---|---|---|---|---|
| **Understanding Only** | | | | | | | |
| LLaVA-v1.5 | 7B | × | 42.8 | 37.7 | 31.4 | 44.7 | 38.23 |
| InternVL2 | 8B | × | 49.0 | 50.1 | 31.9 | 54.5 | 43.47 |
| Med-Flamingo | 8.3B | ✓ | 43.0 | 25.5 | 31.3 | 34.9 | 12.74 |
| LLaVA-Med | 7B | ✓ | 48.1 | 44.8 | 35.7 | 41.3 | 20.54 |
| RadFM | 14B | ✓ | 50.6 | 34.6 | 14.33 | 23.5 | 22.34 |
| HuatuoGPT-Vision-7B | 7B | ✓ | 53.0 | 49.1 | 32.0 | 50.0 | 50.22 |
| MedGemma-4B | 4B | ✓ | 67.6 | 71.2 | 33.7 | 68.4 | 44.0 |
| Lingshu-7B | 7B | ✓ | 62.7 | 77.0 | 59.6 | 82.0 | 52.3 |
| Lingshu-32B | 32B | ✓ | **71.4** | **84.7** | **61.3** | 80.4 | 52.7 |
| GMAI-VL | 7B | ✓ | 66.3 | 72.9 | 39.8 | **88.5** | **61.74** |
| **Unified Understanding and Generation** | | | | | | | |
| Janus | 1.3B | × | 52.8 | 26.9 | 27.9 | 45.7 | 39.30 |
| Bagel | 7B | × | 60.09 | 58.91 | 39.05 | 71.13 | 48.11 |
| HealthGPT-M3 | 3.8B | ✓ | 55.9 | 56.4 | 39.7 | 68.5 | 42.08 |
| HealthGPT-L14 | 14B | ✓ | 58.3 | 64.5 | 44.4 | 74.4 | 43.1 |
| **UniMedVL (Ours)** | 14B | ✓ | 61.9 | 75.4 | 53.5 | 85.8 | 60.75 |

Table 2: Performance comparison of our UniMedVL variants and other baseline models on the text-driven image generation task across different modalities. CS denotes BioMedCLIP Score. **Bold** and underlined text indicate the best performance and second-best performance, respectively.

| Method | CFP | | CXR | | CT | | HIS | | MRI | | OCT | | Ultrasound | | Endoscopy | | Average | |
|---|---|---|---|---|---|---|---|---|---|---|---|---|---|---|---|---|---|---|
| | FID↓ | CS↑ | FID↓ | CS↑ | FID↓ | CS↑ | FID↓ | CS↑ | FID↓ | CS↑ | FID↓ | CS↑ | FID↓ | CS↑ | FID↓ | CS↑ | FID↓ | CS↑ |
| LlamaGen-MediTok | 89.14 | - | **68.16** | - | - | - | 198.63 | - | - | - | - | - | 358.11 | - | - | - | 171.85 | - |
| Bagel | 217.19 | 0.650 | 182.80 | 0.662 | 163.78 | 0.652 | 206.18 | 0.643 | 175.74 | 0.639 | 307.80 | 0.719 | 255.78 | 0.672 | 214.61 | 0.668 | 215.49 | 0.660 |
| UniMedVL-Gen | 77.35 | 0.699 | 190.38 | 0.672 | 79.84 | 0.694 | **107.20** | 0.699 | **82.99** | 0.699 | 107.06 | 0.721 | 100.44 | 0.700 | 121.89 | 0.704 | 108.40 | 0.699 |
| UniMedVL | **53.20** | **0.708** | 73.04 | **0.702** | **73.04** | **0.696** | 149.01 | **0.704** | 90.36 | **0.706** | **99.27** | **0.721** | **95.38** | **0.706** | 133.11 | **0.707** | **96.29** | **0.706** |

## 4.2 PERFORMANCE OF UNIMEDVL

**Medical Visual Understanding Performance**. We evaluate the understanding capabilities of UniMedVL across diverse medical VQA and image comprehension benchmarks. Table 1 presents comprehensive results comparing our model with existing medical VLLMs and unified multimodal models. Table 1 reveals a critical insight: unified architectures can achieve understanding performance comparable to specialized models without sacrificing generation capabilities. UniMedVL demonstrates this principle by maintaining competitive performance across diverse medical domains while supporting both understanding and generation within a single architecture. The key technical insight emerges from comparing unified models (bottom section) with understanding-only models (top section): HealthGPT requires separate model checkpoints for different tasks, while UniMedVL achieves superior performance (85.8% on OmniMedVQA vs. HealthGPT-L14's 74.4%) with seamless task switching. This validates our core contribution that progressive curriculum learning enables effective knowledge sharing between understanding and generation pathways in medical contexts.

**Medical Image Generation Performance.** Table 2 provides empirical evidence for cross-modal knowledge transfer in medical generation. A critical insight emerges from comparing UniMedVL-Gen (generation-only training) with UniMedVL: the average gFID improvement demonstrates that understanding tasks contribute semantic constraints to enhance generation quality. Furthermore, UniMedVL achieves BioMedCLIP scores of 0.706 on average across modalities, indicating strong semantic alignment between generated images and clinical text descriptions. This challenges the conventional assumption that joint training compromises individual task performance, instead showing that medical multimodal learning benefits from task synergy when the data bottleneck is relieved.

**Interleaved Multimodal Tasks Performance.** Table 3 demonstrates UniMedVL's performance on interleaved multimodal tasks. For virtual immunohistochemistry staining (H&E→IHC), UniMedVL achieves 20.27 PSNR, outperforming HealthGPT-M3 by 28%. In MRI super-resolution (4×), our model attains 27.29 PSNR/0.890 SSIM, showing substantial improvement over the unified baseline. For cross-modal synthesis (T2↔FLAIR), UniMedVL reaches 25.07 average PSNR, approaching

Figure 3: **Comprehensive visualization of UniMedVL 's multimodal capabilities.** Demonstration of diverse medical imaging tasks, including text-to-image generation, virtual staining, super resolution, counterfactual generation, and cross-modal synthesis.

specialized models while maintaining unified capabilities. Figure 3 provides qualitative visualization of these capabilities. The key insight emerges from comparing UniMedVL[†] (mixed data training without progressive stages) with full UniMedVL: consistent improvements across all tasks (e.g., 2.16 PSNR gain in virtual staining) demonstrate that our complete progressive training paradigm effectively learns cross-modal relationships that simple fine-tuning cannot capture.

Table 3: **Performance Comparison on specialised generation tasks.** histological staining transformation (H&E to IHC), MRI super-resolution ($4\times$), and medical image translation ($T_2 \leftrightarrow$ FLAIR). PSNR and SSIM are used in medical image translation. † indicates unified fine-tuning variant. **Bold** and underlined text indicate the best performance and second-best performance, respectively.

| H&E→IHC Staining | | MRI Super-Resolution | | Medical Image Translation | | | |
|---|---|---|---|---|---|---|---|
| Method | PSNR/SSIM | Method | PSNR/SSIM | Method | $T_2\rightarrow$FLAIR | FLAIR$\rightarrow T_2$ | Avg |
| CycleGAN | 16.20/0.373 | SRCNN | 28.81/0.892 | ResViT | **24.97**/0.870 | **25.78**/**0.908** | **25.38**/**0.889** |
| Pix2Pix | 18.65/0.419 | VDSR | 30.04/0.914 | pGAN | 24.01/0.864 | 25.09/0.894 | 24.55/0.879 |
| Pix2PixHD | 19.63/0.471 | SwinIR | 31.55/0.933 | pix2pix | 23.15/0.869 | 24.52/0.883 | 23.84/0.876 |
| Pyramid Pix2pix | **21.16**/**0.477** | Restormer | 31.85/0.938 | A-UNet | 23.69/0.873 | 24.56/0.891 | 24.13/0.882 |
| | | AMIR | **31.99**/**0.939** | SAGAN | 24.02/0.860 | 25.10/0.893 | 24.56/0.877 |
| HealthGPT-M3 | 15.81/0.242 | HealthGPT-M3 | 18.37/0.580 | HealthGPT-M3 | 18.88/0.745 | 19.30/0.750 | 19.09/0.748 |
| UniMedVL † | 18.11/0.401 | UniMedVL † | 19.64/0.602 | UniMedVL † | 23.99/0.711 | 23.49/0.732 | 23.74/0.722 |
| **UniMedVL** | 20.27/0.456 | **UniMedVL** | 27.29/0.890 | **UniMedVL** | 24.90/**0.881** | 25.23/0.883 | 25.07/0.882 |

Table 4 evaluates counterfactual generation capabilities with explanatory text. UniMedVL[†] achieves 27.17 gFID and significantly higher text quality metrics (0.2641 BLEU-3, 0.4486 METEOR, 0.4649 ROUGE-L) compared to specialized baselines. The improved counterfactual check rate (0.797 AUROC) demonstrates that our unified training enables generation of clinically plausible scenarios with coherent textual explanations.

Table 4: Comparison of UniMedVL with baseline methods on medical counterfactual generation. **Bold** and underlined text indicate the best performance and second-best performance, respectively.

| Method | Counterfactual Image | | | Explanatory Text | | |
|---|---|---|---|---|---|---|
| | gFID↓ | AUROC↑ | F1↑ | BLEU-3↑ | METEOR↑ | ROUGE-L↑ |
| CXR-IRGen | 35.39 | 0.5236 | 0.7609 | 0.0448 | 0.2115 | 0.1846 |
| ProgEmu | 29.21 | 0.7921 | **0.8914** | 0.1241 | 0.4097 | 0.2606 |
| **UniMedVL** † | **27.17** | **0.7970** | 0.8731 | **0.2641** | **0.4486** | **0.4649** |

Table 5: **Ablation study of the proposed progressive curriculum learning strategy.** UVE refers to the understanding-oriented vision encoder. G and U refer to the generation and understanding subsets of UniMed-5M, respectively. CAG: Caption Augmented Generation, DCOT: Distilled Chain of Thought. **Bold** indicates the best performance and underlined indicates second-best performance.

| Model | UVE | $\mathcal{L}_{\text{NTP}}$ | $\mathcal{L}_{\text{flow}}$ | Data Type | Understanding | | | | Generation | |
| --- | --- | --- | --- | --- | --- | --- | --- | --- | --- | --- |
| | | | | | GMAI-MMBench | SLAKE | PathVQA | OMVQA | gFID↓ | BioMedCLIP↑ |
| **Baseline Comparison** | | | | | | | | | | |
| One-Stage-Joint-Base | × | ✓ | ✓ | U+G | 0.5354 | 0.6560 | 0.4946 | 0.7784 | 123.48 | 0.6945 |
| **Stage 1: Foundation Training** | | | | | | | | | | |
| F-Baseline | × | × | × | - | 0.481 | 0.589 | 0.390 | 0.7113 | 212.73 | 0.662 |
| C-G-only | × | × | ✓ | G | - | - | - | - | 118.5991 | 0.6994 |
| B-U-only | ✓ | ✓ | × | U | 0.505 | 0.5476 | 0.3673 | 0.7723 | - | - |
| H-Joint-Base | ✓ | ✓ | ✓ | U+G | 0.593 | 0.6843 | 0.3649 | 0.8562 | 121.02 | 0.683 |
| **Stage 2: Instruction Tuning** | | | | | | | | | | |
| C-G-only | × | × | ✓ | CAG | - | - | - | - | 108.40 | 0.698 |
| B-U-only | ✓ | ✓ | × | DCOT | 0.5432 | 0.6032 | 0.4526 | 0.8167 | - | - |
| H-Joint-Base | ✓ | ✓ | ✓ | High-quaity U+G | 0.6004 | 0.7418 | 0.5130 | **0.8626** | 120.036 | 0.6989 |
| **Stage 3: Unified Multimodal Training** | | | | | | | | | | |
| H-Joint-Base | ✓ | ✓ | ✓ | Interleaved tasks | **0.6075** | **0.7540** | **0.5346** | 0.8584 | **96.287** | **0.7058** |

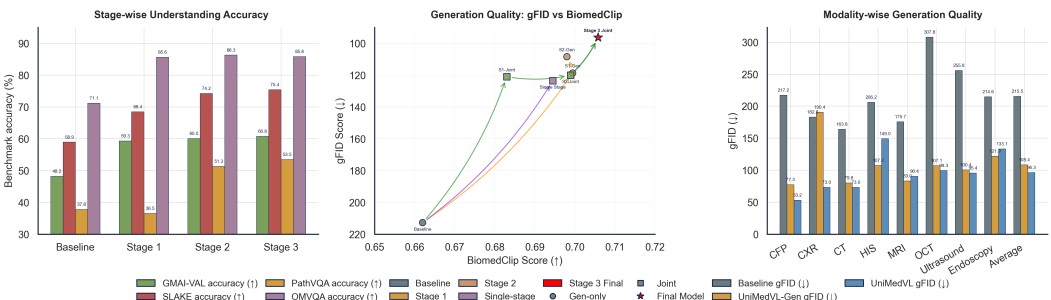

Figure 4: **Visual Comparsion of Performance across different training stages and modalities. (Left:)** Stage-wise understanding accuracy performance. **(Center:)** Generation quality evolution with gFID reduction and BioMedCLIP score enhancement through different training stages. **(Right:)** Modality-specific generation performance comparison across UniMedVL variants.

## 4.3 ABLATION STUDY

Table 5 and Figure 4 validate how our Knowledge level and Analysis level synergistically improve model capabilities. The critical finding is that joint training (H-Joint-Base) consistently outperforms single-task variants during the stage 1, indicating that UniMedVL learns fundamental unified multimodal representations to effectively perform both understanding and generation tasks. The stage 2 further improves the performance on both understanding and generation tasks. The understanding and generation capabilities are enhanced by the instructions with a reasoning process and high-quality image captions, respectively. The stage 3 brings significant improvement, demonstrating that the unified multimodal representations are further improved at this stage and support both understanding and generation tasks.

## 5 CONCLUSION

We presented UniMedVL, a unified framework that simultaneously performs medical image understanding and generation within a single model, validated through extensive experiments on over 5 million medical samples demonstrating both state-of-the-art comprehension (75.40% SLAKE) and competitive generation quality (95.80 gFID). While our current work focuses on 2D medical imaging, the proposed OKA paradigm establishes foundations for exploring diverse medical AI tasks beyond those demonstrated, including 3D volumetric analysis, temporal reasoning, and multimodal clinical decision support. This work represents a critical step toward truly integrated medical AI systems where understanding and generation capabilities synergistically enhance clinical workflows.

ETHICS STATEMENT

This work does not involve experiments on human subjects, patient interventions, or the collection of private medical data. All datasets used in this study are publicly available, including CheXpertPlus, SLAKE, PathVQA, OmniMedVQA, IXI, BraTS 2023, and other open-access medical datasets. Data sources were used in accordance with their respective licenses, and all materials were de-identified prior to use. The purpose of this research is to advance the scientific understanding of unified multimodal modelling for healthcare data rather than to deploy clinical decision-support systems. No patient-level decisions or clinical predictions were made based on model outputs. Expert audits were limited to quality control of publicly available samples and did not involve identifiable patient information.

REPRODUCIBILITY STATEMENT

To ensure reproducibility, all implementation details, model configurations, and training hyperparameters are provided in the Appendix. The full code and configuration files are available in an anonymous repository. The UniMed-5M dataset construction process, including data sources, quality control criteria, and interleaved task synthesis, is fully documented in the Appendix and illustrated in Figure 2. All benchmarks used for evaluation, VQA-RAD, SLAKE, PathVQA, OmniMedVQA, BraTS 2023, and IXI, are publicly available or derived from publicly available sources.

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

# A    APPENDIX

APPENDIX TABLE OF CONTENTS

## A.1 IMPLEMENTATION DETAILS

### A.1.1 TRAINING HYPERPARAMETERS

Table 6: Training hyperparameters and configurations for the three-stage curriculum learning strategy in UniMedVL. These stages collectively implement the Knowledge component of the OKA framework.

| | Stage 1 (Foundation) | Stage 2 (Instruction Tuning) | Stage 3 (Unified Multimodal) |
|---|---|---|---|
| **Hyperparameters** | | | |
| Learning rate | $5 \times 10^{-5}$ | $2.5 \times 10^{-5}$ | $1.0 \times 10^{-5}$ |
| Optimizer | | AdamW | |
| Loss weight (CE : MSE) | | 0.25 : 1.0 | |
| Training steps | 85K | 120K | 70K |
| EMA ratio | | 0.995 | |
| Image Resolution (VAE) | 512-1024 | 512-1024 | 32-1024 |
| Image Resolution (ViT) | 378-980 | 224-518 | 378-980 |
| Max tokens per sample | 18.5K | 20K | 27K |
| Dropout | | Text: 0.3, ViT/VAE: 0.05 | |
| ViT training | Trainable | Frozen | Frozen |
| VAE training | | Frozen | |
| Understanding branch | | Trainable | |
| LLM training | | Trainable | |
| **Data Sampling Ratio (%)** | | | |
| Text-Only | 5 | 5 | 3 |
| Text-to-Image (T2I) | 25 | 45 | 35 |
| Image-to-Text (I2T) | 75 | 40 | 37 |
| Interleaved | - | 10 | 25 |

**Detailed Training Strategy Implementation.** Our training employs a three-stage curriculum learning approach that implements the Knowledge component within the OKA framework. We use the AdamW optimizer throughout all stages:

- Stage 1 (Foundation Training) establishes basic medical understanding over 85K steps with a learning rate of $5 \times 10^{-5}$. The data composition prioritizes image-to-text tasks (75%), complemented by text-to-image generation (25%) and pure text data (5%). This stage trains both ViT and LLM components end-to-end while keeping the VAE frozen. The image resolution is restricted with the range from 512-1024 pixels for the generation branch and 378-980 pixels for the understanding branch.

- Stage 2 (Instruction Tuning) extends training to 120K steps with a reduced learning rate of $2.5 \times 10^{-5}$. The data mixture evolves to balance text-to-image (45%) and image-to-text (40%) tasks, while introducing interleaved multimodal datasets (10%). The ViT encoder is frozen at this stage to preserve learned visual features. Token capacity increases to 20K per sample.

- Stage 3 (Unified Multimodal Training) focuses on interleaved generation capabilities over 70K steps with a learning rate of $1.0 \times 10^{-5}$. This stage significantly increases interleaved dataset usage (25%) while maintaining balanced generation (35%) and understanding (37%) tasks. The expanded token budget (27K) and broader image resolution range (32-1024 pixels for generation) support interleaved tasks, including medical image super-resolution, modality translation, and counterfactual generation.

**Hardware Requirements and Training Infrastructure.** Our model training was conducted using 8× A800 GPUs (80GB memory each) for experimental validation. However, for optimal training efficiency and to fully exploit the model's capacity, we recommend a minimum configuration of 16× A800 GPUs or equivalent hardware.

**Technical Implementation Details.** The training employs a unified loss function that balances understanding and generation objectives with a CE:MSE weight ratio of 0.25:1.0. We apply consistent

dropout rates across all stages (Text: 0.3, ViT/VAE: 0.05) to prevent overfitting. The EMA coefficient is set to 0.995 for stable model convergence. Throughout training, the VAE remains frozen to maintain stable latent representations.

**Rationale for Using Pretrained VAE without Fine-tuning.** Our approach leverages a general-purpose pretrained VAE model from FLUX (Black Forest Labs, 2024) without medical domain-specific fine-tuning. This design choice addresses two core questions: (1) the reconstruction capability of pretrained VAE on medical imaging modalities, and (2) the cost-benefit trade-off of fine-tuning versus preserving existing capabilities. Regarding the first question, we conducted comprehensive reconstruction experiments across eight medical imaging modalities to evaluate performance. For the second question, considering that our training data is not specifically designed for reconstruction optimization, we did not pursue domain-specific fine-tuning to avoid potential degradation of the model's general-purpose capabilities while maintaining stable latent representations throughout our progressive training stages.

Table 7: Reconstruction quality evaluation of pretrained VAE models on medical imaging modalities.

| Metric | Model | $f_d$ | CFP | CT | CXR | Endoscopy | HIS | MRI | OCT | Ultrasound |
|---|---|---|---|---|---|---|---|---|---|---|
| **rFID (Lower is Better)** | | | | | | | | | | |
| | VAE (FLUX) | 8 | 13.22 | 5.81 | 5.42 | 11.77 | 10.00 | 10.58 | 13.23 | 9.64 |
| | Direct End-to-end VAE (FLUX) | 8 | 14.05 | 30.59 | 23.28 | 39.56 | 44.64 | 37.95 | 17.33 | 31.58 |
| | VQGAN | 8 | 27.22 | 15.97 | 33.57 | 27.73 | 21.33 | 67.68 | 29.48 | 18.66 |
| | Emu3-VQ | 8 | 16.27 | 11.83 | 27.91 | 20.83 | 13.52 | 69.89 | 25.43 | 11.99 |
| | MedITok | 16 | 14.39 | 7.88 | 22.27 | 10.66 | 6.32 | 46.54 | 17.64 | 6.55 |
| **PSNR (Higher is Better)** | | | | | | | | | | |
| | VAE (FLUX) | 8 | 34.58 | 37.34 | 37.09 | 35.33 | 34.50 | 34.30 | 34.58 | 33.59 |
| | Direct End-to-end VAE (FLUX) | 8 | 35.11 | 34.43 | 31.28 | 31.98 | 29.69 | 34.82 | 30.83 | 35.17 |
| | VQGAN | 8 | 35.40 | 31.13 | 29.28 | 25.60 | 29.54 | 20.94 | 24.79 | 31.68 |
| | Emu3-VQ | 8 | 28.96 | 36.11 | 31.68 | 28.96 | 34.32 | 22.08 | 27.57 | 35.81 |
| | MedITok | 16 | 37.72 | 36.32 | 31.69 | 29.17 | 23.55 | 23.55 | 25.49 | 34.42 |
| **SSIM (Higher is Better)** | | | | | | | | | | |
| | VAE (FLUX) | 8 | 0.892 | 0.951 | 0.973 | 0.934 | 0.922 | 0.921 | 0.892 | 0.938 |
| | Direct End-to-end VAE (FLUX) | 8 | 0.842 | 0.848 | 0.904 | 0.900 | 0.938 | 0.934 | 0.867 | 0.816 |
| | VQGAN | 8 | 0.923 | 0.885 | 0.753 | 0.768 | 0.844 | 0.484 | 0.248 | 0.317 |
| | Emu3-VQ | 8 | 0.943 | 0.928 | 0.793 | 0.847 | 0.957 | 0.547 | 0.751 | 0.955 |
| | MedITok | 16 | 0.953 | 0.937 | 0.855 | 0.890 | 0.972 | 0.660 | 0.935 | 0.883 |

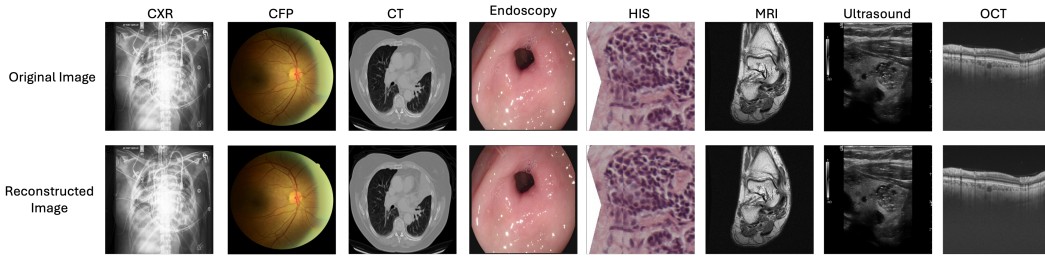

Figure 5: **Qualitative comparison of VAE reconstruction quality across diverse medical imaging modalities.** Visual examples demonstrating reconstruction fidelity across eight medical imaging modalities (CFP, CT, CXR, Endoscopy, HIS, MRI, OCT, Ultrasound) using the pretrained FLUX VAE without domain-specific fine-tuning.

The empirical evaluation demonstrates that the VAE (FLUX) achieves competitive reconstruction performance across eight distinct medical imaging modalities without requiring domain-specific fine-tuning. With a compression factor of $f_d = 8$, the model consistently delivers low rFID scores, competitive PSNR values, and robust SSIM scores. To evaluate the necessity of domain-specific adaptation, we performed direct end-to-end fine-tuning of the FLUX VAE on medical imaging data (highlighted in red in Table 7). The empirical results demonstrate that domain-specific fine-tuning yields negligible performance improvements for generation tasks across medical modalities, exhibiting inconsistent variations in rFID scores with marginal changes in corresponding metrics. Consequently,

these observations validate the deployment of pretrained VAE models without domain-specific fine-tuning for 2D medical imaging generation applications within our experimental framework.

### A.1.2 RECONSTRUCTION FIDELITY FOR CLINICALLY IMPORTANT SMALL LESIONS

While Table 7 demonstrates competitive aggregate reconstruction metrics (rFID, PSNR, SSIM) across diverse medical modalities, these metrics may not fully capture the preservation of small but clinically critical structures such as polyps in endoscopy, dermatoscopic features in skin lesions, or fractures in radiographs. To address this concern, we conducted targeted qualitative analysis focusing on the reconstruction fidelity of fine anatomical details and pathological findings that are essential for clinical diagnosis.

We selected representative cases from three imaging modalities where small lesion detection is clinically critical: (1) an endoscopy image containing polyp, (2) a dermoscopy image with skin lesion with globules, and (3) a x-ray image with fractures. For each case, we compared the original image with its VAE-reconstructed counterpart, examining both full-field views and magnified regions of interest (ROIs) centered on the lesions.

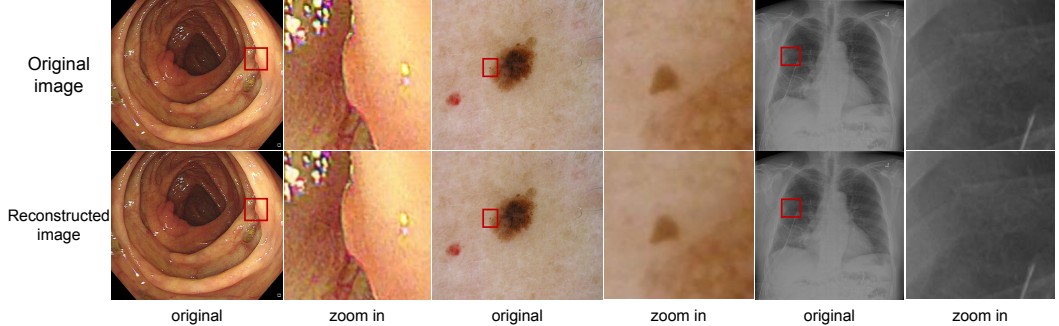

Figure 6: **Preservation of clinically important small lesions in VAE reconstruction.** Side-by-side comparison of original images (top row) and VAE-reconstructed images (bottom row) across three medical imaging scenarios. **Left:** Endoscopy image showing a polyp (highlighted in red box). **Middle:** Dermoscopy image displaying skin lesion with globules (highlighted in red box). **Right:** X-ray image with fractures (highlighted in red box). **Original** denotes the original image, and **Zoom-in** denotes the zoomed-in view of the lesion within the red boxes. The magnified views demonstrate that the FLUX VAE preserves fine structural details essential for clinical interpretation, despite being a general-purpose encoder not specifically fine-tuned for medical imaging.

Figure 6 illustrates that the pretrained FLUX VAE maintains visually discernible fidelity for small pathological features. In the endoscopy image, the polyp's morphology and surface texture remain well preserved in the reconstruction, with boundary definition comparable to the original. For the dermoscopy image, the characteristic globular patterns, key diagnostic features for distinguishing benign nevi from melanoma—are clearly visible in both the original and reconstructed versions. In the X-ray image, the highlighted region and surrounding anatomical structures maintain structural coherence post-reconstruction. These qualitative observations suggest that general-purpose VAE also preserves clinically relevant fine-grained details that are critical for downstream diagnostic tasks within our unified framework.

## A.2 DATASET STATISTICS

### A.2.1 DATASET COMPOSITION DETAILS

Table 8: Overview of training stage data distribution, showing data composition, task types, and scale statistics across different stages. Stage 2 utilised the high-quality subset of stage 1 datasets.

| Training Stage | Total Entries | Task Categories |
|---|---|---|
| **Stage 1: Foundation Training** | | |
| Understanding Tasks | 4.0M | Image comprehension, VQA |
| Generation Tasks | 1.6M | Text-to-image, controllable generation |
| *Stage 1 Subtotal* | *5.6M* | *Foundation capabilities* |
| **Stage 2: Instruction Tuning** | | |
| Understanding Tasks | 698K | Image CoT, clinical reasoning |
| Generation Tasks | 668K | Enhanced T2I, medical translation |
| CoT Understanding | 317K | Chain-of-thought reasoning |
| Text-only Tasks | 230K | Medical QA, clinical dialogue |
| *Stage 2 Subtotal* | *1.9M* | *Knowledge integration* |
| **Stage 3: Unified Multimodal Training.** | | |
| Interleaved Tasks | 330K | 5 interleaved tasks |
| *Stage 3 Subtotal* | *0.33M* | *Unified capabilities* |
| **Total Dataset** | **5.6M** | **All medical tasks** |

### A.2.2 MEDICAL DOMAIN AND MODALITY DISTRIBUTION

Table 9: Major datasets detailed information, showing key dataset contributions sorted by data volume. For open-source datasets, the reported numbers indicate the actual subset sizes used in our training pipeline after filtering.

| Dataset Name | Total Entries | Primary Tasks |
|---|---|---|
| PMC-OA (Lin et al., 2023) | 1.0M | Text-to-Image Generation |
| Quilt-1m (Ikezogwo et al., 2023) | 644K | Histopathology Understanding |
| Healthgpt (Lin et al., 2025) | 638K | Clinical Reasoning, Image Caption |
| PubMedVision (Chen et al., 2024a) | 385K | Controllable T2I Generation |
| Gmai-vl (Li et al., 2024) | 288K | Enhanced T2I Generation |
| Bigbio (Fries et al., 2022) | 262K | Clinical Reasoning with CoT |
| CheXpertPlus (Chambon et al., 2024) | 223K | Medical Report Understanding |
| PMC VQA (Zhang et al., 2023b) | 204K | Image Caption |
| Internvl (Chen et al., 2024c) | 188K | Disease Classification, Clinical Reasoning |
| Medicat (Subramanian et al., 2020) | 132K | Controllable T2I Generation |
| Medical-diff-vqa (Hu et al., 2023) | 129K | Image Caption, Entity Recognition |
| PMC-Inline (Wu et al., 2025a) | 121K | Multi-image Understanding |
| IXI T2/T1 SR 4x (Information eXtraction from Images (IXI) Project, 2024) | 161K | Super resolution |
| BraTS23 Modality Tran (Baltruschat et al., 2023) | 52K | Cross modal synthesis |
| SynthRAD Brain (MR to CT/CT to MR) (Thummerer et al., 2025) | 66K | Cross modal synthesis |
| SynthRAD Pelvis (MR to CT/CT to MR) (Thummerer et al., 2025) | 42K | Cross modal synthesis |
| ICG-CXR dataset (Ma et al., 2025b) | 10K | Counterfactual generation |
| BCI dataset (Liu et al., 2022a) | 5K | Virtual immunohistochemistry staining |
| **Total (Selected Datasets)** | **4.55M** | – |
| **Others Datasets** | **1.05M** | – |
| **Grand Total** | **5.6M** | **All Tasks** |

### A.2.3 MODALITY AND ANATOMY DISTRIBUTION

Figure 7 illustrates the comprehensive statistics of our curated medical datasets, showing both modality distribution and anatomical coverage. These statistics highlight the diversity and quality of data used across different training stages.

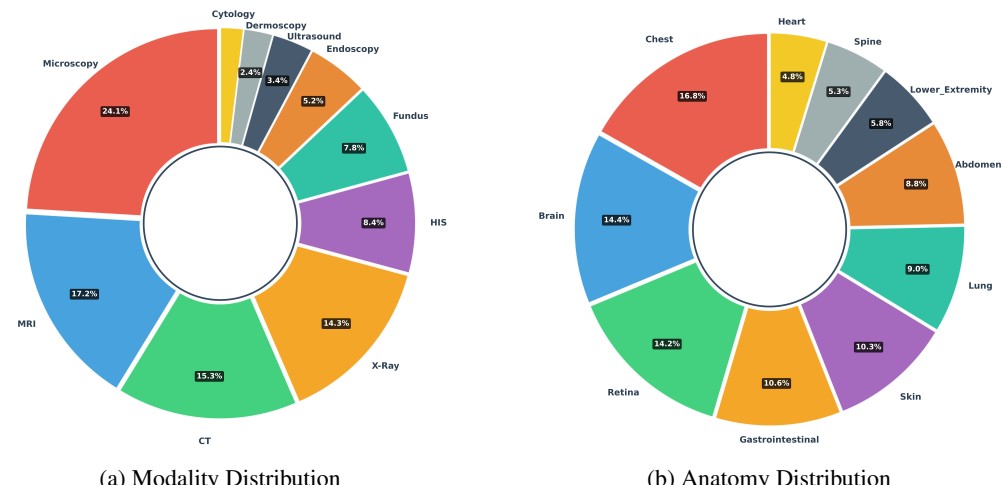

(a) Modality Distribution                    (b) Anatomy Distribution

Figure 7: Comprehensive statistics of our curated medical datasets, showing modality distribution and anatomy distribution. These statistics demonstrate the diversity and quality of data used across different training stages.

### A.3 DATA ENHANCEMENT PIPELINE: CAG IMPLEMENTATION

This section presents the complete prompt templates used in our Caption Augmented Generation (CAG) pipeline for image generation tasks, as described in Section 3. The CAG pipeline consists of two main stages: (1) structured medical description generation for quality control, and (2) caption fusion that combines original captions with generated descriptions.

#### A.3.1 STAGE 1: STRUCTURED DESCRIPTION GENERATION

---

**Stage 1: Structured Description Generation Prompt**

**Purpose:** Generate four-level structured medical image descriptions for quality control and similarity computation

```
You are a universally expert medical image analyst, proficient in all
imaging modalities and anatomical systems.
Your input is a single medical image, with no supplementary information.
Your only task is to provide a comprehensive, objective, and structured
description at four distinct levels, from the highest overview down to
the most specific and exceptional findings.
You must not offer any diagnostic, interpretive, or clinical advice.

---

Output Structure (Four-Level, Top-to-Bottom -- definitions for your
internal guidance; do NOT reproduce these headings in your answer)

LEVEL 1: IMAGE TYPE & GLOBAL CONTEXT
· In one sentence, state the presumed imaging modality (if visually
  clear), main body region(s), and overall image category (e.g.,
  cross-sectional, projectional, histological).
· Example: "This is an axial CT image of the abdomen and pelvis,
  showing cross-sectional anatomy at the level of the lower kidneys."

LEVEL 2: MACRO-ANATOMICAL OVERVIEW
· In 2-4 concise lines, summarize the global distribution and layout
  of major anatomical regions, dominant structures, and any clearly
  visible large-scale abnormalities, masses, or disease patterns.
· Describe anatomical orientation, symmetry, major organ relationships,
  and other visually prominent features.

LEVEL 3: ORGAN / SUBREGION DETAILS -- must be the most detailed section
· In 6-12 lines (use complete sentences), describe the visual
  appearance of individual organs, vessels, bones, or other relevant
  subregions.
· Provide precise, granular, reproducible details so that all main
  features can be reconstructed.
· Maintain strict objectivity; do not include diagnostic language.

LEVEL 4: SPECIAL OR INCIDENTAL FINDINGS
· List any unusual devices, postsurgical changes, image artifacts,
  rare morphologic features, or observations not already mentioned above.
· If none are visible, explicitly state: "No distinct pathological
  or incidental findings are visible."

Writing Instructions
1. Write the entire description as one continuous paragraph that
   implicitly follows the LEVEL 1 → LEVEL 4 order--do not include
   level headings, bullet points, or numbered lists in the paragraph.
2. Do not use bullet points elsewhere (except within the examples).
3. For more complex images, the portion corresponding to LEVEL 3 should
   naturally be longer; for simpler cases, keep it proportionally concise.
4. Avoid any clinical judgement or speculation--describe only what is
   directly visible.
```

#### A.3.2 STAGE 2: CAPTION FUSION ENHANCEMENT

This stage fuses original captions with Stage 1 generated structured descriptions to create enhanced descriptions for image generation tasks.

---

**Stage 2: Caption Fusion Enhancement Prompt**

**Purpose:** Fuse original captions with structured descriptions for enhanced image generation prompts

```
You are a universally expert medical image analyst, proficient in all
imaging modalities and anatomical systems.

CRITICAL CONSTRAINT: You must maintain absolute anatomical consistency.
NEVER change, assume, or modify the anatomical location described in the
original caption. Do not make assumptions about different anatomical locations or
transfer descriptions between different body parts.

Your input consists of:
1. A structured, objective, four-level description derived from a locally
   deployed AI model (following a strict hierarchy from global overview
   to specific findings).
2. An original, data-derived textual description containing high-density,
   potentially diagnostic or interpretative information, which may lack
   structured clarity.

Your task is to:
• First, critically review and confirm the completeness of the structured
  description generated by the local model.
• Then, systematically extract and objectively incorporate relevant,
  visually verifiable details from the original data-derived description,
  enhancing information density without including diagnostic, interpretive,
  or clinical judgement.
• Clearly indicate and explicitly include visually evident anatomical
  abnormalities, structural deviations, or incidental observations present
  in the original data but omitted in the structured description.

Output Structure (Four-Level, Top-to-Bottom)
LEVEL 1: IMAGE TYPE & GLOBAL CONTEXT
• In one sentence, state the presumed imaging modality, main body
  region(s), and overall image category.

LEVEL 2: MACRO-ANATOMICAL OVERVIEW
• In 2--4 concise lines, summarize global anatomical distribution,
  dominant structures, anatomical symmetry or deviations, and clearly
  visible large-scale abnormalities.

LEVEL 3: ORGAN / SUBREGION DETAILS -- must be the most detailed section
• In 6--12 complete sentences, describe individual organs, bones,
  vessels, and other relevant anatomical subregions in precise,
  reproducible detail.
• Objectively highlight visually confirmed abnormalities or structural
  deviations derived from the original data description.

LEVEL 4: SPECIAL OR INCIDENTAL FINDINGS
• Explicitly mention unusual devices, postsurgical changes, rare
  morphological features, or visually detectable anomalies present in
  the original description yet absent in the structured description.
• Clearly state the absence of commonly expected baseline anatomical
  or pathological features if definitively not observed in the image.

Writing Instructions
1. Write the final enhanced description as a single, continuous paragraph
   implicitly following LEVEL 1 → LEVEL 4 order--do not include explicit
   level headings, bullet points, or numbered lists.
2. Avoid any clinical judgement, diagnostic language, or speculative
   interpretation--include only details directly verifiable from visual
   inspection.
3. Start your output with "Please generate a realistic [modality] image
   showing" to make it a proper generation instruction.
```

### A.3.3 STAGE 3: THINKING-ENHANCED RESPONSE GENERATION

This stage aims to elicit the reasoning process from the medical foundation model (MediGama-27B-IT) by prompting it to explicitly generate its internal thinking steps. We leverage this specialized medical model to simulate detailed reasoning processes through the structured prompt format. The resulting data, which includes both the explicit thinking traces and the final responses, is then used to train our model.

---

**Stage 3: Thinking-Enhanced Response Generation Prompt (Revised v2)**

**Purpose:** Generate medical image responses with thinking tags for enhanced reasoning and quality control

```
System: You are a medical image generator. You create [modality] images based
on clinical descriptions. Your responses should describe what features you
have generated in the image from the creator's perspective. Use bullet points
to organize the anatomical structures and clinical features you have included
in your generated image.

User: Based on this clinical description: "[clinical_description]"

You have been given the corresponding medical image. Please provide a response
following this format:

Required format:
<think>Analyzing the clinical description, I need to generate an image that
captures: 1) The key pathological process described, 2) The anatomical
structures involved, 3) The specific imaging characteristics for [modality].
Based on the clinical presentation, I should include [key features reasoning].
[structured_caption if available]</think>

Here/This is the generated [modality] image that displays:
• [anatomical structure or clinical finding 1]
• [anatomical structure or clinical finding 2]
• [anatomical structure or clinical finding 3]

IMPORTANT:
1. In the <think> tag, reason through WHAT you need to generate and WHY based
   on medical knowledge
2. Respond from the GENERATOR perspective – describe what features you have
   CREATED/GENERATED in the image
3. Use the exact format above with bullet points (•) to list features
4. Start with 'Here is the generated [modality] image that displays:'
5. Each bullet point should describe a specific anatomical structure,
   clinical finding, or visual feature that you have included
6. Do NOT use observational language like 'shows', 'visible', 'can be seen'
   – instead use generative language like 'displays', 'includes',
   'features', 'contains'

Note: The thinking tag should reflect your decision-making process: "I need
to generate X because Y", "The clinical description indicates I should
include Z", etc.
```

The enhanced captions from Stage 2 (if the process "generating" is not generated successfully) and Stage 3 (if the process "thinking" is generated successfully) are sampled and then submitted to the Expert Review system (Section A.5) for final validation.

## A.4 DOWNSTREAM TASK RESULTS

### A.4.1 MEDICAL REPORT GENERATION

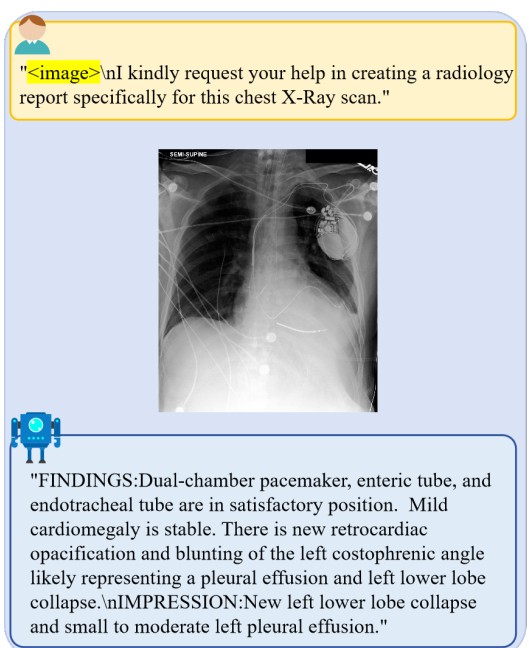

Figure 8: Medical report generation examples showing input images and generated reports.

### A.4.2 VISUAL QUESTION ANSWERING

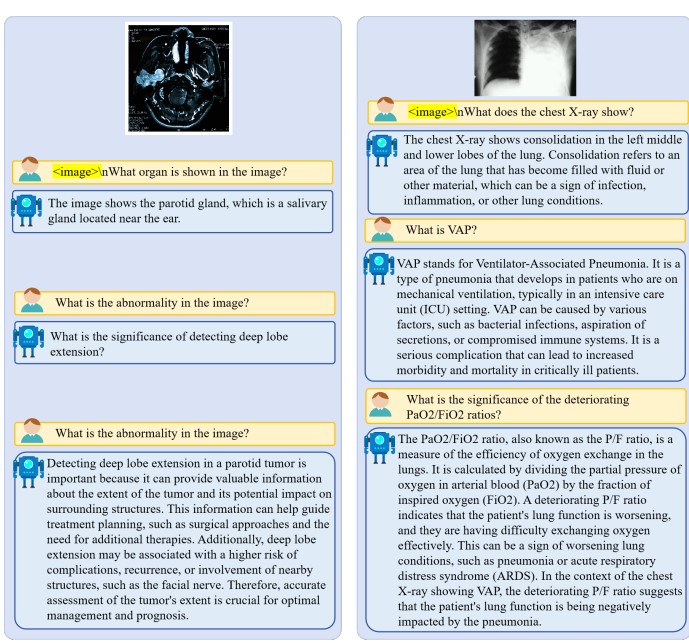

Figure 9: Visual question answering examples across different medical imaging modalities.

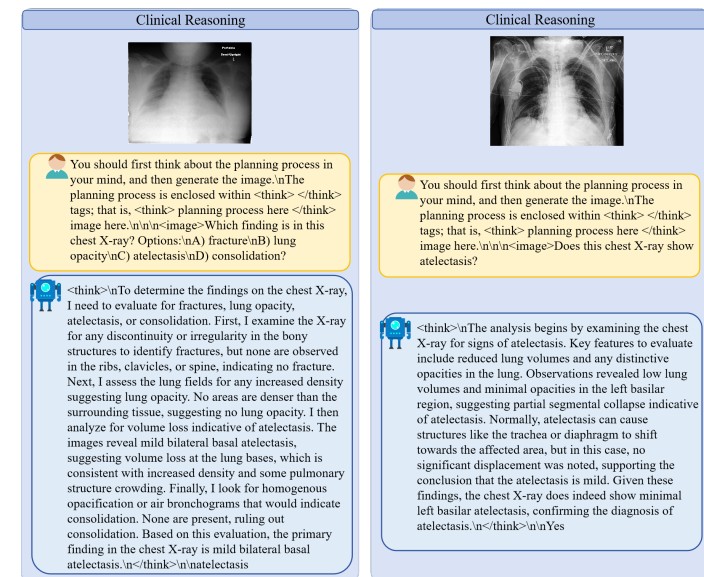

Figure 10: Visual question answering examples across different medical imaging modalities(part 2).

### A.4.3 MEDICAL IMAGE GENERATION

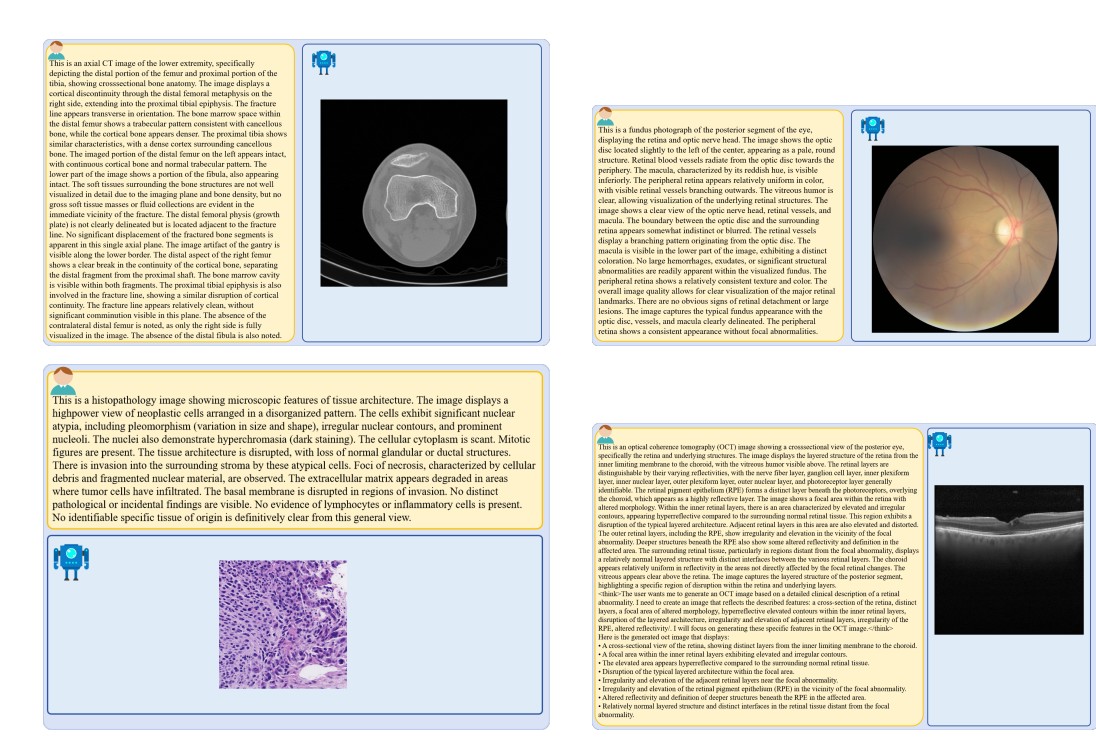

Figure 11: Medical image generation examples with text prompts.

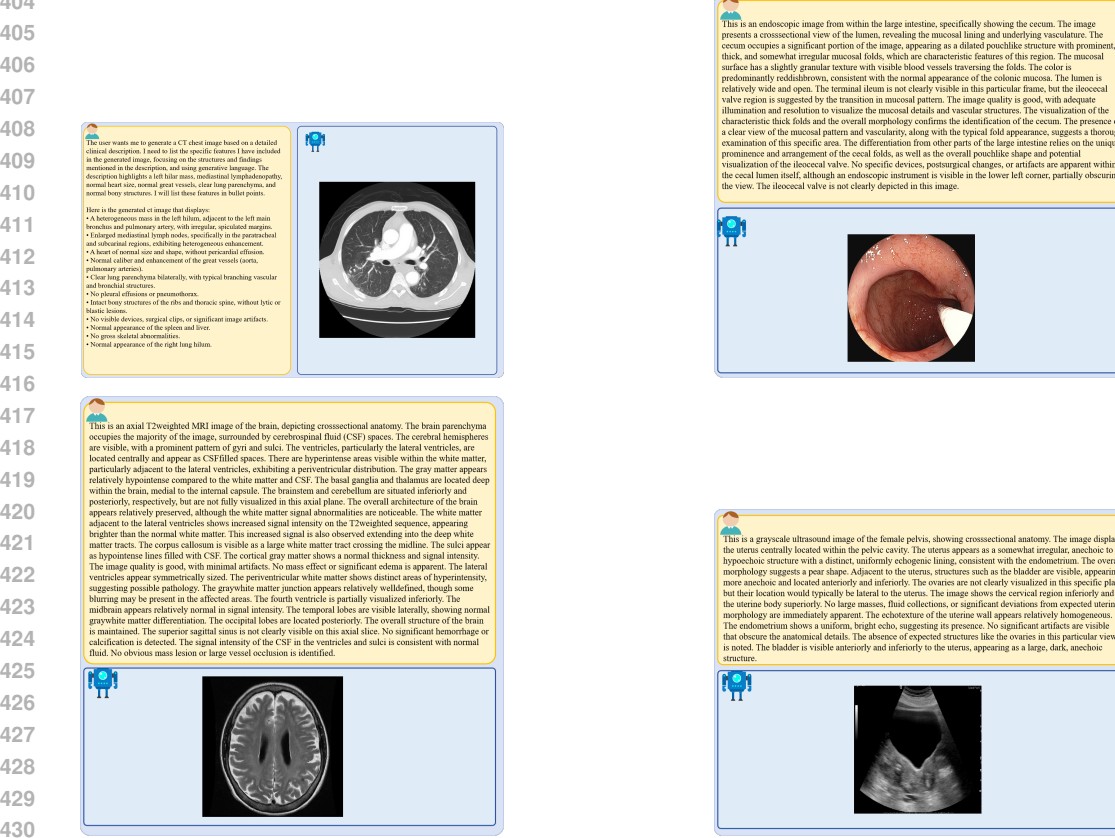

Figure 12: Medical image generation examples with text prompts(part 2).

### A.4.4 INTERLEAVED TASKS

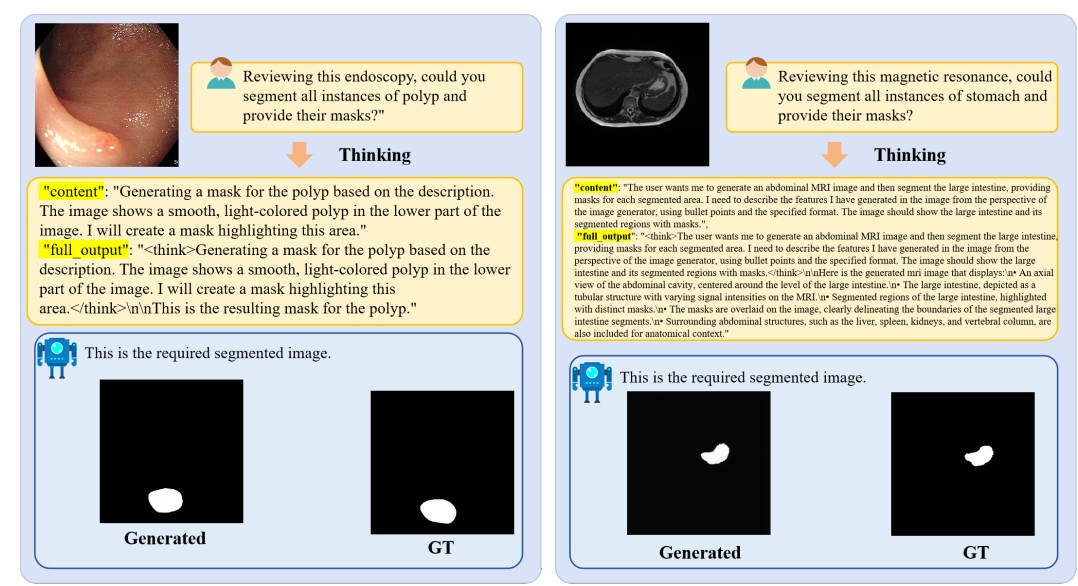

Figure 13: Medical Image Prompt Segmentation.

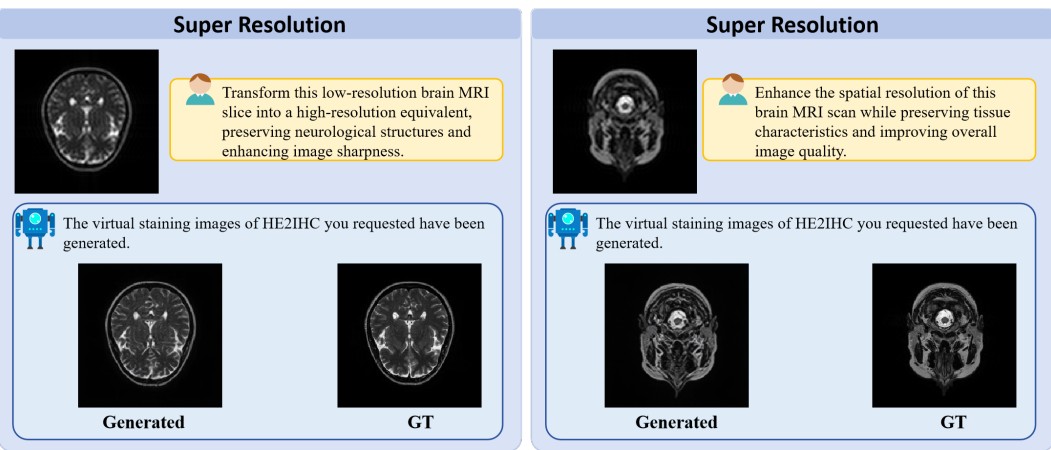

Figure 14: Super Resolution.

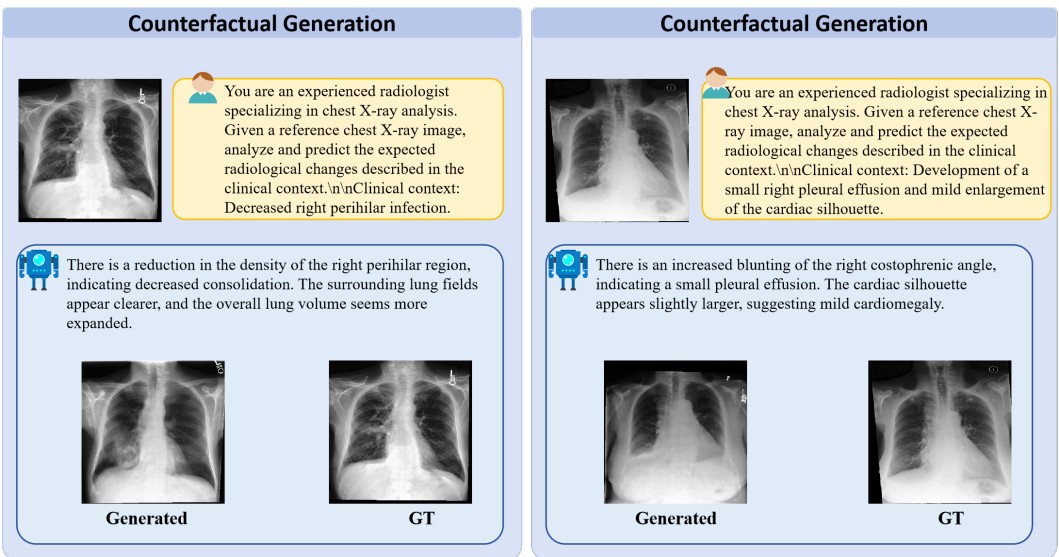

Figure 15: Counterfactual Generation.

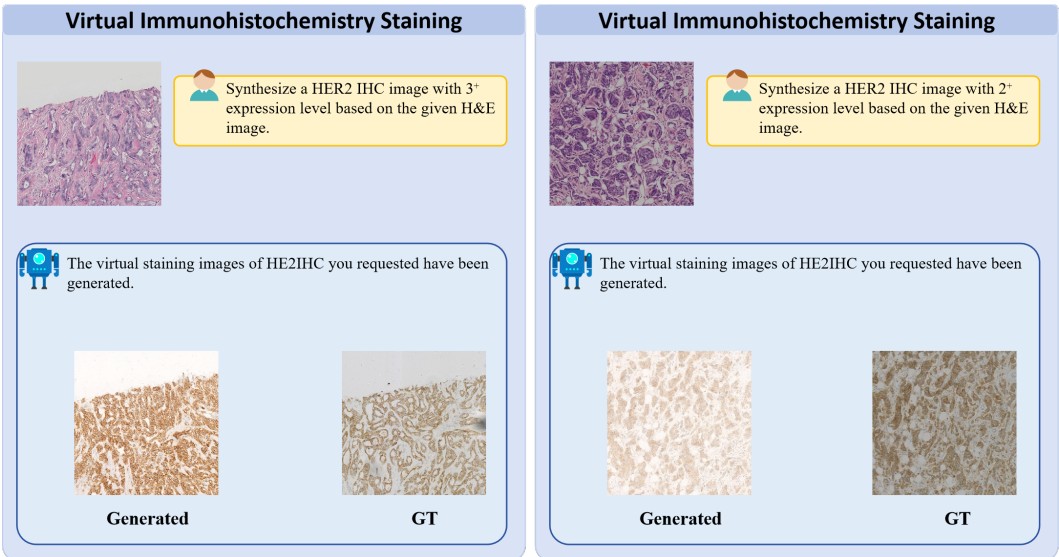

Figure 16: Virtual Immunohistochemistry Staining.

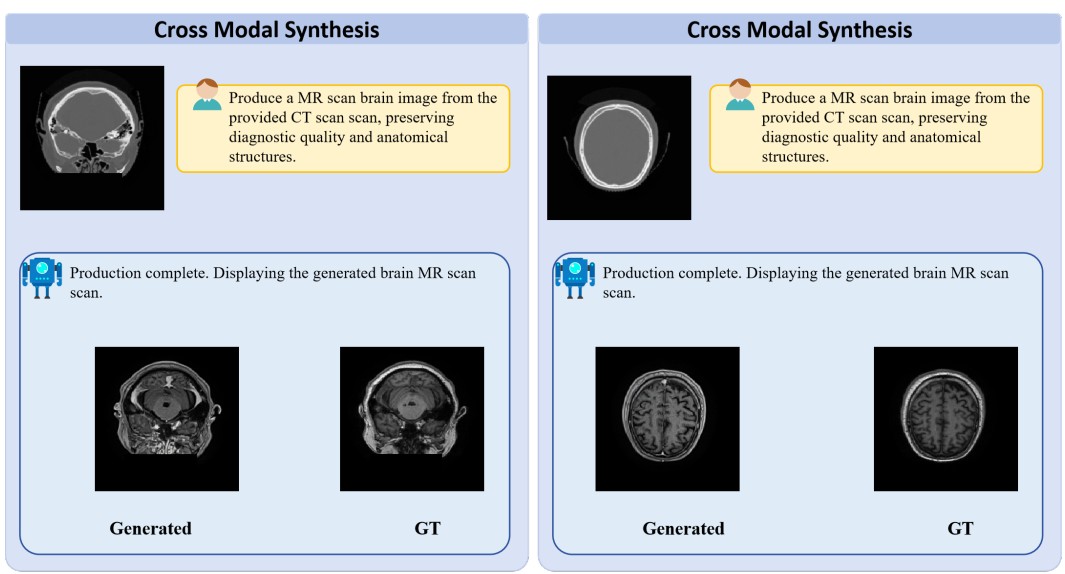

Figure 17: Cross Modal Synthesis.

### A.5 EXPERT REVIEW VALIDATION SYSTEM

This section presents an expert review validation system that evaluates the quality of our UniMed-5M dataset construction and two caption generation approaches described in the Data Enhancement Pipeline (Section A.3):

**Simple approach:** Caption fusion that combines structured descriptions from Stage 1 with original captions (Stage 2 of CAG pipeline).

**Thinking-enhanced approach:** Incorporates an additional planning process with <think> tags that integrates reasoning steps before medical image generation (Stage 3 of CAG pipeline). The validation system evaluates both data quality and methodological effectiveness.

#### A.5.1 EXPERT REVIEW FRAMEWORK OVERVIEW

Our expert review validation system is designed around a seven-dimensional medical evaluation framework that assesses medical AI performance.

Our evaluation framework encompasses seven dimensions that assess the synthetic quality of medical image captions. The framework begins with **Modality Match (0-1)**, which measures consistency between images and declared medical imaging modalities, followed by **Factual Accuracy (0-5)** that evaluates the precision of anatomical structure and pathological finding descriptions. **Information Completeness (0-5)** assesses coverage of diagnostically relevant key information, while **Position/Quantity Accuracy (0-5)** measures precision in anatomical localization and quantitative assessments. The framework also incorporates **Professionalism (0-5)** to evaluate adherence to medical reporting standards, **Planning Coherence (0-5)** to assess systematic thinking and logical organization quality, and finally **Clinical Reasoning (Turing Test) (0-5)** to measure approximation to human expert-level performance.

**Expert Validation Protocol:** Experts conducted audits of 200 samples across all seven dimensions. The evaluation process achieved inter-rater agreement exceeding 0.85 across all dimensions.

#### A.5.2 EVALUATION DIMENSION ANALYSIS

Figure 18 presents the correlation analysis and comparative results. Figure 18a shows inter-dimensional correlations, while Figure 18b compares the two generation approaches.

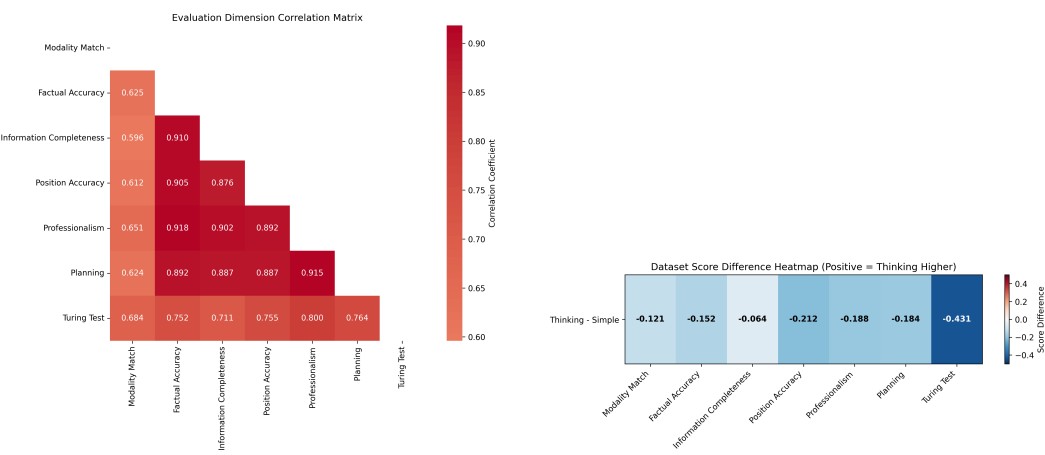

(a) Correlation matrix between evaluation dimensions.  (b) Score difference heatmap comparing thinking and simple approaches.

Figure 18: **Expert evaluation analysis.** (a) Correlation matrix revealing inter-dimensional relationships (Pearson correlation coefficients ranging from 0.60 to 0.92). (b) Score difference heatmap comparing thinking and simple approaches (negative values indicate simple approach scores higher; all dimensions scored on 0-5 scale except Modality Match on 0-1 scale).

### A.5.3 DATASET QUALITY COMPARISON ANALYSIS

Figure 19 compares the two generation approaches across all evaluation dimensions. The radar chart (Figure 19a) shows closely aligned performance profiles.

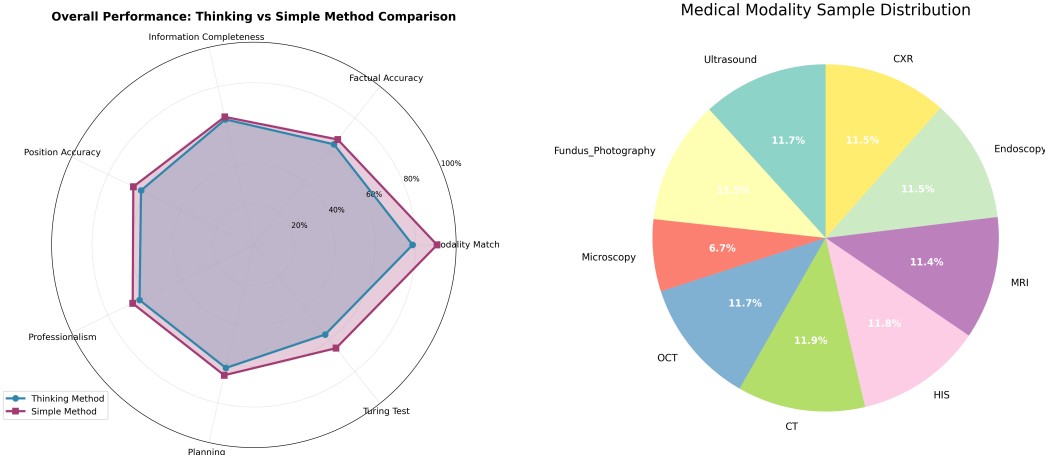

(a) Performance comparison: Thinking vs Simple approaches across evaluation dimensions.

(b) Medical imaging modalities distribution

Figure 19: **Expert validation overview.** (a) Radar chart comparing performance profiles of thinking and simple approaches across all seven evaluation dimensions. (b) Pie chart showing balanced representation across medical imaging modalities, ensuring comprehensive coverage.

### A.5.4 MEDICAL MODALITY-SPECIFIC ANALYSIS

Figure 20 presents modality-specific performance across nine medical imaging modalities. Figure 20a shows statistical comparisons, and Figure 20b displays detailed performance metrics.

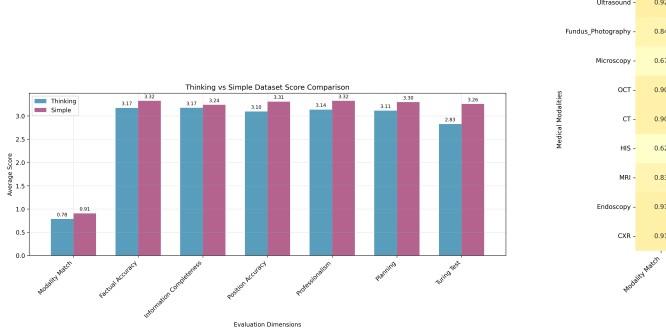
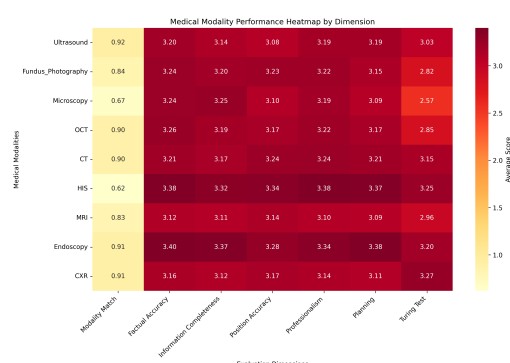

(a) Statistical comparison between thinking and simple approaches.

(b) Modality-specific performance analysis.

Figure 20: **Comprehensive performance analysis.** (a) Bar chart showing mean scores with confidence intervals. (b) Heatmap displaying modality-specific performance scores.

## A.6 DOWNSTREAM TASK PERFORMANCE

### A.6.1 MEDICAL REPORT GENERATION

Table 10: **Medical report generation performance on MIMIC-CXR dataset.** Evaluation of automated radiology report generation using three metrics: ROUGE-L (lexical similarity), RaTE (radiology-specific terminology accuracy), and RadCliQ$^{-1}$ (clinical quality assessment). Higher scores indicate better performance for all metrics. Baseline results are sourced from Xu et al. (2025a). **Bold** indicates best performance and underlined indicates second-best performance.

| Models | MIMIC-CXR ROUGE-L | MIMIC-CXR RaTE | MIMIC-CXR RadCliQ$^{-1}$ |
|---|---|---|---|
| GPT-4.1 | 9.0 | 51.3 | 57.1 |
| Claude Sonnet 4 | 20.0 | 45.6 | 53.4 |
| Gemini-2.5-Flash | 25.4 | 50.3 | 59.4 |
| Med-R1-2B | 19.3 | 40.6 | 42.4 |
| MedLM-R1-2B | 20.3 | 41.6 | 48.3 |
| MedGemma-8B-IT | 25.6 | **52.4** | 62.9 |
| LLaVA-Med-7B | 15.0 | 12.8 | 52.9 |
| HuatuoGPT-V-7B | 23.4 | 48.9 | 48.2 |
| BioMediX2-8B | 20.0 | 44.4 | 53.0 |
| Qwen2.5VL-7B | 24.1 | 47.0 | 55.1 |
| InternVL2-8B | 23.2 | 47.0 | 56.2 |
| InternVL3-8B | 22.9 | 48.2 | 55.1 |
| Lingshu-7B | **30.8** | 52.1 | **69.2** |
| HealthGPT-14B | 21.4 | 48.4 | 52.7 |
| HuatuoGPT-V-34B | 23.5 | 48.5 | 47.1 |
| MedDr-40B | 15.7 | 45.2 | 47.0 |
| InternVL3-14B | 22.0 | 48.6 | 46.5 |
| Qwen2.5VL-32B | 15.7 | 47.5 | 45.2 |
| InternVL2.5-38B | 22.7 | 47.5 | 54.9 |
| InternVL3-38B | 22.8 | 47.9 | 47.2 |
| Lingshu-32B | 28.8 | 50.8 | 67.1 |
| UniMedVL | 19.2 | 45.0 | 42.4 |

### A.6.2 CXR LUNG OPACITY IMAGE TRANSLATION

Table 11: **Unpaired chest X-ray opacity removal translation performance.** Task involves transforming CXRs with lung opacities to opacity-free counterparts while preserving anatomical structures. Evaluation metrics: FID (Fréchet Inception Distance) and KID (Kernel Inception Distance), where lower values indicate better image quality and distributional similarity. Baseline results are sourced from published literature. **Bold** indicates best performance and underlined indicates second-best performance.

| Model | FID $\downarrow$ | KID $\downarrow$ |
|---|---|---|
| **Baselines** | | |
| Original CXRs | 81.80 | 0.043 |
| Munit | 109.4 | 0.073 |
| Unit | 103.2 | 0.061 |
| CycleGAN | 208.3 | 0.216 |
| Uvcgan | 210.4 | 0.225 |
| Drit | 117.6 | 0.087 |
| AAMA-CDA | 67.18 | 0.016 |
| **Unified Models** | | |
| HealthGPT-M3 | 62.19 | 0.031 |
| **UniMedVL**[†] | **35.1** | **0.008** |

### A.7    FAILURE CASES AND ANALYSIS

We present representative failure cases of UniMedVL organized by task type: medical image generation, medical image editing, and medical image understanding, with illustrative examples shown in Figures 21 and 22.

#### A.7.1    MEDICAL IMAGE GENERATION:

**Global appearance and background artefacts.**    Although UniMedVL generally produces realistic images across modalities, a characteristic failure mode in text-to-image generation concerns embedded text and annotations (Figure 21). In some synthesised samples, the model hallucinates spurious on-image text or renders partially legible words, labels, or font styles that do not appear in the corresponding real clinical images or do not match typical acquisition overlays. These artefacts do not alter the main anatomical content but introduce visually unnatural patterns. As shown in Figure 21, the generated chest X-ray exhibits spurious text overlays (red boxes highlight the artefacts), the ultrasound image contains hallucinated text labels that deviate from standard clinical annotations, and the CT scan shows partially corrupted metadata text at the bottom that does not match typical DICOM overlay formatting.

#### A.7.2    MEDICAL IMAGE EDITING:

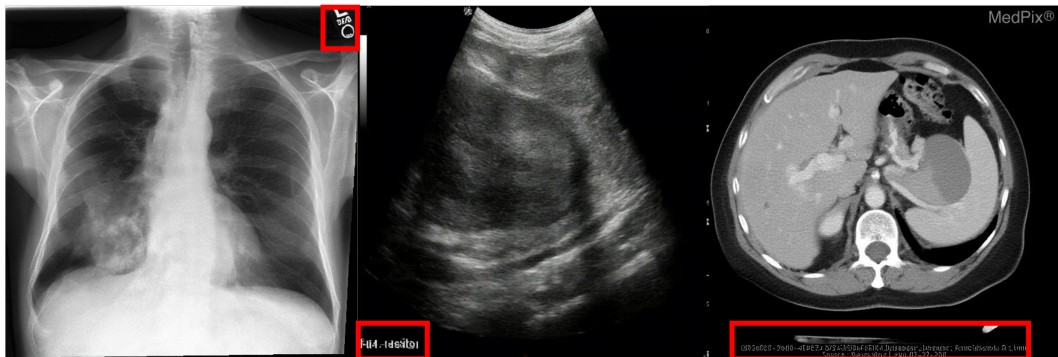

Figure 21: **Text and annotation artefacts in medical image generation.** Representative examples in the medical image generation task showing hallucinated or corrupted text elements across different imaging modalities. **Left:** Chest X-ray with spurious text overlay in the upper region (red box). **Center:** Ultrasound image displaying hallucinated text labels that do not conform to standard clinical annotation conventions (red box). **Right:** CT scan showing partially corrupted metadata text at the bottom edge that deviates from typical DICOM overlay formatting (red box).

**Structure preservation in generation and editing.**    For interleaved editing-style tasks (e.g., virtual staining, super-resolution, cross-modal synthesis, counterfactual generation), UniMedVL does not always perfectly preserve all spatial structures outside the region being semantically edited (Figure 22). For instance, in some counterfactual CXR generations, small devices or lines (e.g., catheters) can become slightly blurred or shifted, even when the main pathological change is correctly applied. Figure 22 illustrates these challenges across three representative cases: in the brain MRI cross-modal synthesis tasks (top two rows), the generated images show subtle structural discrepancies in the cerebellar and temporal regions compared to ground truth, and in the chest X-ray counterfactual generation (bottom row), while the model successfully modifies the target pathological region, occasionally minor shifts and blurring are observable in the cardiac silhouette boundaries.

## Input    Ground Truth    Generated Image

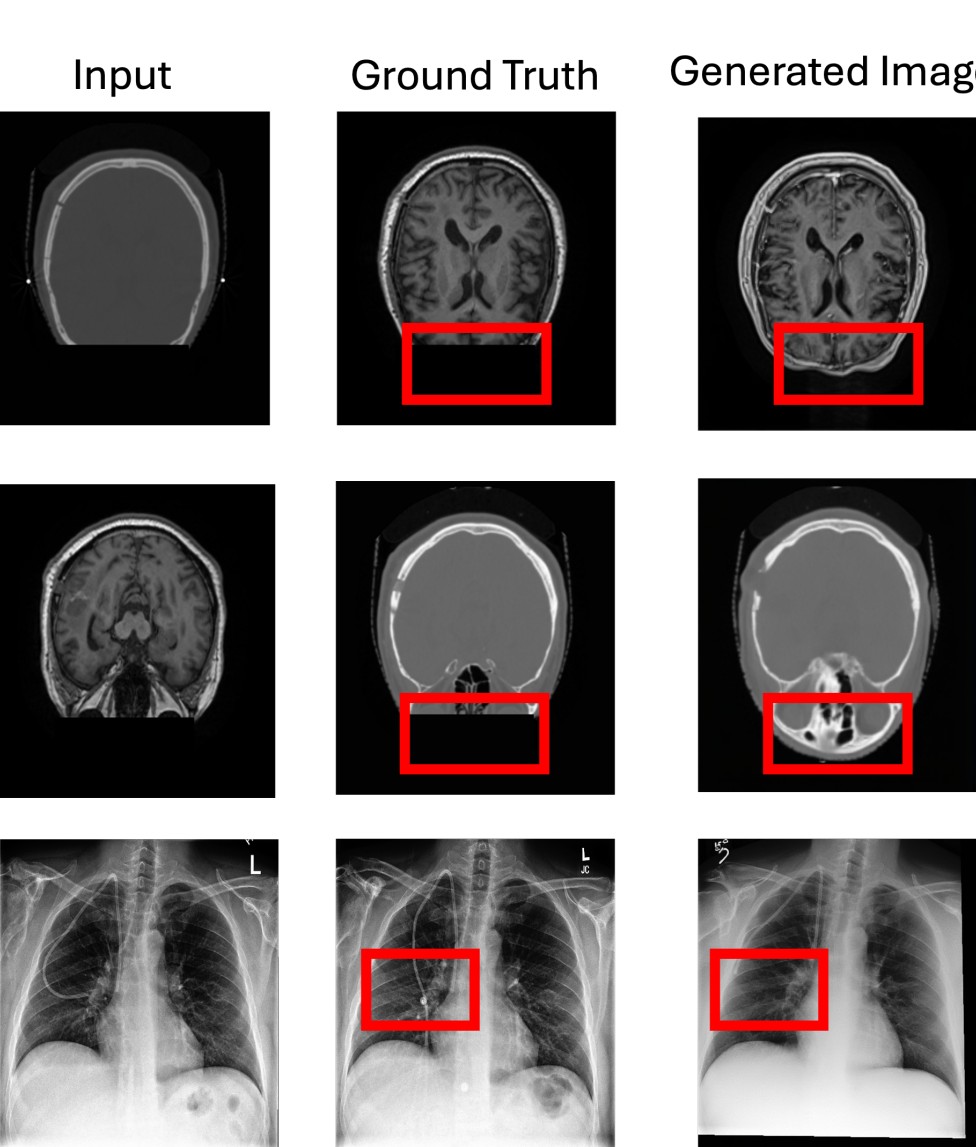

Figure 22: **Structural preservation challenges in interleaved generation and editing tasks.** Comparative analysis in the medical image editing task across different medical image translation scenarios. Each row presents a triplet of **Input**, **Ground Truth**, and **Generated Image**. **Top row:** Brain MRI cross-modal synthesis, where the generated image exhibits subtle structural distortions in the cerebellar region compared to the ground truth. **Middle row:** Reverse brain MRI synthesis shows minor misalignment in the temporal lobe structures. **Bottom row:** Chest X-ray counterfactual generation task demonstrating reduction of pleural effusion; while the target pathological modification is applied, the generated image shows slight blurring and positional shifts in the cardiac silhouette and mediastinal borders.

### A.7.3 MEDICAL IMAGE UNDERSTANDING:

Table 12: **UniMedVL performance on GMAI-MMBench validation set.** Accuracy across 18 clinical VQA sub-categories.

| Task Category | Accuracy |
|---|---|
| Overall | 0.607 |
| Attribute Recognition | 0.659 |
| Blood Vessels Recognition | 0.593 |
| Bone | 0.623 |
| Cell Recognition | 0.513 |
| **Counting** | **0.457** |
| Disease Diagnosis | 0.669 |
| **Image Quality Grading** | **0.440** |
| Microorganism Recognition | 0.793 |
| Muscle | 0.580 |
| Nervous Tissue | 0.925 |
| Organ Recognition – Abdomen | 0.657 |
| Organ Recognition – Head and Neck | 0.845 |
| Organ Recognition – Pelvic | 0.560 |
| Organ Recognition – Thorax | 0.747 |
| **Severity Grading** | **0.372** |
| **Surgeon Action Recognition** | **0.287** |
| **Surgical Instrument Recognition** | **0.310** |
| **Surgical Workflow Recognition** | **0.471** |

**Medical demanding quantitative and fine-grained recognition scenarios.** In medical demanding tasks that require precise quantitative reasoning or fine-grained visual recognition, the unified model shows clear limitations across multiple task categories. Table **??** presents UniMedVL's performance on GMAI-MMBench validation set, revealing particularly low accuracy on surgical video recognition tasks: Surgeon Action Recognition, Surgical Instrument Recognition, and Surgical Workflow Recognition. The model also struggles with quantitative assessment tasks including Counting, Image Quality Grading, and Severity Grading.

**Multimodal Medical Quantitative Assessment**

From the image input, please count and tell me the number of lymphocytes shown.
A: "the number of lymphocytes:30"
B: "the number of lymphocytes:7"
C: "the number of lymphocytes:24"
D: "the number of lymphocytes:25"

On this immunohistochemical slide, lymphocytes correspond to the brown-stained cells against blue nuclei. Visually estimating, I count ~30 such brown cells across the field. Therefore, among the options provided, I select **A**: "the number of lymphocytes:30" as the best match to the observed staining pattern.

The error arises from coarse visual estimation: clusters of overlapping brown staining are treated as separate lymphocytes, inflating the count. The model fails to delineate individual nuclei and to exclude weak or nonspecific staining, leading to overcounting versus the true value of 24 lymphocytes (option C).

Figure 23: **Counting failure in lymphocyte quantification. Green indicates the correct answer, and red indicates our model's prediction.**

**Surgeon Action Recognition**

Focusing on the boxed section of this endoscopy video, what surgical action is in progress?
A: "bladder neck dissection"
B: "pulling seminal vesicle"
C: "bagging prostate"
D: "sucking smoke"

Based on the boxed endoscopic view, I compare visual cues with typical urologic steps. Bladder-neck dissection usually dominates such frames, whereas seminal-vesicle traction, prostate bagging, or smoke suction require more specific devices or anatomy not clearly visible. Given the limited context, I infer the action is bladder-neck dissection **(A)**.

This answer reflects procedural misclassification. The model relies on generic priors ("typical" frames of bladder-neck dissection) instead of carefully inspecting the boxed region for the retrieval bag and specimen, which are diagnostic of prostate bagging. It underutilizes the explicit prompt to recognize the specific surgical step in progress.

Figure 24: **Surgeon Action Recognition failure. Green indicates the correct answer, and red indicates our model's prediction.**

**Surgical Instrument Recognition**

Looking at the region delineated by a rectangle in the provided endoscopy image, which option would be the most target-like?
    A: "bipolar"
    B: "scissors"
    C: "hook"
    D: "irrigator"

Within the boxed endoscopic region, I compare the visible instrument with typical laparoscopic tools. The elongated tubular appearance suggests a fluid-delivery device rather than a cutting tool or curved hook. Thus, among the options "bipolar," "scissors," "hook," and "irrigator," I judge the instrument most consistent with an irrigator **(D)**

The mistake stems from coarse shape-based reasoning. The model focuses on the shaft's cylindrical form and ignores the distal jaws, which clearly resemble scissor blades rather than an irrigation nozzle. By not attending to fine-grained tip morphology and articulation, it misclassifies the laparoscopic scissors (correct answer B) as an irrigator.

Figure 25: **Surgical Instrument Recognition failure. Green indicates the correct answer, and red indicates our model's prediction.**

## Surgical Workflow Recognition

Please identify which of the following options the surgical procedure shown in the image belongs to.
    A: "durotomy"
    B: "nasal corridor creation"
    C: "haemostasis"
    D: "anterior sphenoidotomy"

Endoscopic view shows nasal/sinus cavity with instrument working on bony/soft tissue surface. This appearance is more consistent with an endonasal approach to the sphenoid sinus than with durotomy or generic nasal corridor creation or simple haemostatic manoeuvres. Thus I classify it as anterior sphenoidotomy **(D)**.

The model overemphasizes coarse anatomical context (endonasal access, apparent sinus wall) and underutilizes dynamic cues that the instrument is primarily controlling bleeding rather than creating a bony opening. Consequently, a supportive step (haemostasis, C) is misinterpreted as a definitive sinus procedure (anterior sphenoidotomy, D)

Figure 26: **Surgical Workflow Recognition failure. Green indicates the correct answer, and red indicates our model's prediction.**

**Image Quality Grading**

Assessing the boxed area in this endoscopy image, which artifact stands out?
A: "instrument artifacts"
B: "saturation"
C: "blur"
D: "low contrast"

In the boxed region, a sharply bright, featureless patch is visible on the mucosal surface. I interpret this intense highlight as glare caused by the endoscope or another tool, i.e., an instrument-related artifact, rather than blur, low contrast, or general image-quality degradation. Therefore I select option A, "instrument artifacts."

The model conflates extreme brightness with instrument glare, ignoring that no tool contour is visible and that the signal is locally clipped with complete texture loss. These properties are typical of sensor saturation rather than reflection from an instrument surface, so the correct dominant artifact is saturation (option B), not instrument artifacts.

Figure 27: **Image Quality Grading failure. Green indicates the correct answer, and red indicates our model's prediction.**

**Severity Grading**

Please determine the severity of the provided image according to the core-needle biopsy grading protocol.
A: "axillary lymph node (aln) metastasis - n0)"
B: "axillary lymph node (aln) metastasis - n+(1-2)"
C: "axillary lymph node (aln) metastasis - n+(>2)"

In this core-needle biopsy image, I see numerous metastatic tumor cells filling much of the field. According to the aln grading scheme (n0: none; n+(1–2): limited; n+(>2): extensive), this density appears compatible with involvement beyond two high-power fields, so I classify it as n+(>2) **(C)**.

The error arises from misusing the HPF-based protocol: a single densely infiltrated patch is overinterpreted as implying involvement of >2 fields. The model extrapolates tumor extent beyond the visible image instead of restricting grading to sampled HPFs, so the burden actually corresponds to n+(1–2) metastasis (correct answer B).

Figure 28: **Severity Grading failure. Green indicates the correct answer, and red indicates our model's prediction.**

## A.8 EFFICIENCY ANALYSIS

We measure inference efficiency with single expert activation. Comparison with BLIP3-o, Janus, and HealthGPT is provided in Tables 13.

**Image generation (Table 13(a)).** UniMedVL (14B) requires 40.03 TFLOPs per image and 28.39 GB peak memory, whereas BLIP3-o 8B requires 142.59 TFLOPs per image and 25.55 GB. This corresponds to approximately 3.6x lower compute per image for UniMedVL with only about 11% higher peak memory (28.39 vs. 25.55 GB), even though UniMedVL uses a dual-encoder and MoT design and has almost twice the parameters. UniMedVL's compute cost is also close to that of Janus 7.42B (35.56 TFLOPs per image) while providing a larger unified model.

**VQA on GMAI-MMBench (Table 13(b)).** UniMedVL achieves 25.86 tokens/s with 2.256 TFLOPs per sample and 28.25 GB peak memory, compared with BLIP3-o's 30.40 tokens/s, 9.307 TFLOPs per sample, and 18.21 GB. Thus, UniMedVL attains an approximately 4.1x reduction in FLOPs per sample with comparable throughput (about 85% of BLIP3-o's tokens/s), at the cost of higher peak memory due to the dual-encoder design. Compared to another 14B unified model, HealthGPT-L14 (12.57 tokens/s, 3.009 TFLOPs, 29.22 GB), UniMedVL is roughly twice as fast in throughput and more compute-efficient.

Table 13: **Efficiency Evaluation.** Comparison of throughput and computational costs across unified medical multimodal models with batch size 1.

**(a) Image Generation Throughput**

*Warm up with 10 images and measure efficiency over 20 images*

| Model | Parameters | FLOPs/Image (TFLOPs) | Peak Mem (GB) |
|---|---|---|---|
| Janus | 1B | 10.01 | 5.19 |
| HealthGPT-M3 | 3.8B | 15.22 | 10.23 |
| Janus | 7.42B | 35.56 | 17.10 |
| BLIP3-o | 8B | 142.59 | 25.55 |
| **UniMedVL** | **14B** | **40.03** | **28.39** |

**(b) VQA Understanding Throughput (GMAI-MMBench validation set)**

*Warm up with 50 questions and measure efficiency over 150 VQA questions*

| Model | Parameters | Tokens/s | FLOPs/Sample (TFLOPs) | Peak Mem (GB) |
|---|---|---|---|---|
| Janus | 1B | 70.11 | 0.498 | 4.46 |
| HealthGPT-M3 | 3.8B | 22.13 | 1.304 | 8.79 |
| Janus | 7.42B | 52.94 | 1.894 | 14.59 |
| BLIP3-o | 8B | 30.40 | 9.307 | 18.21 |
| **UniMedVL** | **14B** | **25.86** | **2.256** | **28.25** |
| HealthGPT-L14 | 14B | 12.57 | 3.009 | 29.22 |

### A.9 STATEMENT ON THE USE OF LARGE LANGUAGE MODELS

Throughout the preparation of this manuscript, we utilized large language models to enhance the quality of the text. Specifically, we employed GPT and Claude for tasks related to language refinement, including correcting grammar and spelling, improving sentence clarity, and ensuring a consistent academic tone.

The core scientific contributions, including the formulation of the problem, the proposed methodology, the design and execution of experiments, and the interpretation of results, are entirely the work of the authors. All text generated or modified by the LLM was meticulously reviewed, edited, and revised by the authors to ensure it accurately reflects our original ideas and findings. The authors bear full and final responsibility for all content presented in this paper.

