# OpenReview forum: "UniMedVL: Unifying Medical Multimodal Understanding and Generation through Observation-Knowledge-Analysis"
_ICLR.cc/2026/Conference — ICLR 2026 Conference Withdrawn Submission_

### Official Review · Reviewer_8GAa · 2025-10-20

**Soundness:** 3
**Presentation:** 2
**Contribution:** 3
**Rating:** 6
**Confidence:** 3

**Summary:**

The paper introduces the UniMedVL framework, which aims to achieve unified modeling of medical understanding and generation tasks through the Observation–Knowledge–Analysis (OKA) paradigm. Its core ideas include: 1. Constructing the UniMed-5M dataset, which covers multiple medical imaging modalities; 2. Employing Progressive Curriculum Learning—comprising *Foundation Training*, *Instruction Tuning*, and *Unified Multimodal Training*—to enable deep semantic space fusion across modalities and tasks; 3. Designing the unified model UniMedVL, which utilizes MoT architecture to establish a shared latent space for both image understanding and generation. Experimental results demonstrate that UniMedVL surpasses existing unified models in average accuracy across five medical understanding benchmarks (VQA-RAD, SLAKE, PathVQA, OmniMedVQA, and GMAI-MMBench), and achieves performance on multiple generation tasks that matches or approaches that of task-specific models.

**Strengths:**

1. Accurate problem formulation and well-motivated study: The authors clearly identify the fragmentation of current medical MLLMs across tasks and modalities, focusing their goal on building a unified framework for understanding and generation. Experiments effectively validate the synergistic benefits between understanding and generation in medical-grade tasks.

2. The UniMed-5M dataset shows strong potential: Its three-stage quality control process (Coarse Filtering → Medical Alignment → Expert Validation) enhances data usability and cross-modal consistency. Moreover, the dataset integrates multiple modalities and task types; if released, it could greatly support research on unified frameworks for medical MLLMs.

3. Complete training paradigm: Progressive Curriculum Learning enables a smooth transition from semantic understanding to generative capability, and Table 5 together with Figure 4 effectively demonstrate the synergy between this training paradigm and the MoT architecture.

4. Comprehensive experimental setup.

**Weaknesses:**

1. Difference from Bagel: UniMedVL largely inherits the architectural design of Bagel, raising the question of whether the model sufficiently accounts for the additional differentiation required by the medical domain—particularly regarding data heterogeneity and domain-specific characteristics that differ from general-domain multimodal models.

2. Limited innovation in the multi-stage training paradigm: Several existing works (e.g., Lingshu, HealthGPT) have already explored the benefits of multi-stage training for medical reasoning. Progressive, task-granular post-training strategies are becoming standard practice, and UniMed-5M does not appear to introduce conceptual innovations beyond engineering refinements.

3. Missing comparison with state-of-the-art medical MLLMs: The main tables lack direct comparisons with leading models such as Lingshu and MedGemma, which would strengthen the empirical credibility of the claims.

4. Failure case analysis: A detailed investigation of failure cases is necessary to identify the boundary conditions or sample types under which the model underperforms or fails.

5. Writing and structure could be improved: The manuscript contains numerous long and compound sentences, making it less readable; a clearer and more concise presentation would improve accessibility.

**Questions:**

1. The dataset shows imbalance across modalities (as illustrated in Figure 6). Have the authors experimented with modality-balanced sampling or loss reweighting to mitigate this issue?

2. In the medical report generation task (Appendix Table 10), UniMedVL underperforms compared to Lingshu-7B. Is this due to differences in task optimization objectives? Could multi-task loss rebalancing improve the model’s performance on this task?

3. Do the authors plan to release part of the UniMed-5M dataset or model inference weights to facilitate reproducibility and further research within the community?

4. Please also refer to the weaknesses section for additional points that would benefit from clarification or further discussion.

---

> ### Author Response · Authors · 2025-11-22
> **Response to Weakness 1**
>
> ## Weakness 1.
>
> > 1. Difference from Bagel: UniMedVL largely inherits the architectural design of Bagel, raising the question of whether the model sufficiently accounts for the additional differentiation required by the medical domain—particularly regarding data heterogeneity and domain-specific characteristics that differ from general-domain multimodal models.
>
> A1. Thank you for raising this important point about differentiation from Bagel. While UniMedVL reuses the backbone of Bagel, our contributions are explicitly tailored to the medical domain along both the data and model development axes as follows:
>
> (i) **On the data/task side,** we construct UniMed-5M as a medical-specific corpus that unifies medical image understanding, image generation, and a broad set of clinically motivated interleaved tasks (e.g., counterfactual imaging with textual explanations) into a unified multimodal input format.
>
> (ii) **On the model development side,** our Observation–Knowledge–Analysis (OKA) framework is aligned with clinical workflows: Observation builds the unified medical corpus, i.e., UniMed-5M; Knowledge introduces capabilities via a progressive curriculum learning strategy (from foundation training to instruction tuning to unified multimodal training); and Analysis develops UniMedVL to simultaneously solve medical image understanding, generation, and interleaved tasks.
>
> Together, these medical-domain-aware designs go beyond a direct Bagel transplant and are key to adapting the unified model architecture to heterogeneous medical data and diverse medical tasks.

---

> ### Author Response · Authors · 2025-11-22
> **Response to Weakness 2**
>
> ## Weakness 2.
>
> > 2. Limited innovation in the multi-stage training paradigm: Several existing works (e.g., Lingshu, HealthGPT) have already explored the benefits of multi-stage training for medical reasoning. Progressive, task-granular post-training strategies are becoming standard practice, and UniMed-5M does not appear to introduce conceptual innovations beyond engineering refinements.
>
> A2. Thank you for raising this concern about the novelty of our progressive curriculum learning paradigm. We clarify the differences as follows. Prior medical MLLMs such as Lingshu and HealthGPT essentially adopt a two-stage paradigm—continual multimodal pretraining followed by instruction tuning (with HealthGPT’s “second stage” mainly adapting LoRA modules). In contrast, our Progressive Curriculum Learning explicitly introduces a third stage of unified multimodal training on the formulated interleaved tasks. This stage further improves the performance of both image understanding and image generation, as shown in Table 5. The results indicate that UniMedVL develops medical multimodal understanding and generation capabilities, meaning that its image generation capability can promote its image understanding capability, and vice versa, during this training stage. Thus, our approach conceptually goes beyond traditional continual post-training strategies.

---

> ### Author Response · Authors · 2025-11-22
> **Response to Weakness 3**
>
> ## Weakness 3.
>
> > 3. Missing comparison with state-of-the-art medical MLLMs: The main tables lack direct comparisons with leading models such as Lingshu and MedGemma, which would strengthen the empirical credibility of the claims.
>
> A3.  Thank you for pointing out the missing comparison with recent medical MLLMs. We have added the comparison with Lingshu and MedGemma **in Table 1 in the manuscript**. These results show that UniMedVL outperforms MedGemma-4B on most benchmarks and achieves comparable performance with Lingshu-7B. UniMedVL also outperforms Lingshu-32B on OmniMedVQA and GMAI-MMBench with fewer activated parameters, where UniMedVL activates 7B parameters and Lingshu-32B activates 32B parameters. UniMedVL demonstrates highly competitive visual understanding performance compared with specialized understanding-focused models such as Lingshu and MedGemma, and also solves image generation and interleaved tasks such as counterfactual generation.
>
> ***Table 1**: Comparison of UniMedVL with other LVLMs and unified multi-modal models on medical visual understanding tasks. **Bold** and * values indicate the best and second-best performance, respectively.*
>
> | **Model**               | **Params** | **Medical** | **VQA-RAD** | **SLAKE** | **PathVQA** | **OmniMedVQA** | **GMAI-MMBench** |
> |-------------------------|-----------:|:-----------:|------------:|----------:|------------:|---------------:|-----------------:|
> | **Understanding Only**  |            |             |             |           |             |                |                  |
> | LLaVA-v1.5              | 7B         | ✗           | 42.8        | 37.7      | 31.4        | 44.7           | 38.23            |
> | InternVL2               | 8B         | ✗           | 49.0        | 50.1      | 31.9        | 54.5           | 43.47            |
> | Med-Flamingo            | 8.3B       | ✓           | 43.0        | 25.5      | 31.3        | 34.9           | 12.74            |
> | LLaVA-Med               | 7B         | ✓           | 48.1        | 44.8      | 35.7        | 41.3           | 20.54            |
> | RadFM                   | 14B        | ✓           | 50.6        | 34.6      | 14.33       | 23.5           | 22.34            |
> | HuatuoGPT-Vision-7B     | 7B         | ✓           | 53.0        | 49.1      | 32.0        | 50.0           | 50.22            |
> | MedGemma-4B             | 4B         | ✓           | 67.6*      | 71.2      | 33.7        | 68.4           | 44.0             |
> | Lingshu-7B              | 7B         | ✓           | 62.7        | 77.0*    | 59.6*      | 82.0           | 52.3             |
> | Lingshu-32B             | 32B        | ✓           | **71.4**    | **84.7**  | **61.3**    | 80.4           | 52.7             |
> | GMAI-VL                 | 7B         | ✓           | 66.3        | 72.9      | 39.8        | **88.5**       | **61.74**        |
> | **Unified Understanding and Generation** | |        |             |           |             |                |                  |
> | Janus                   | 1.3B       | ✗           | 52.8        | 26.9      | 27.9        | 45.7           | 39.30            |
> | Bagel                   | 7B         | ✗           | 60.09       | 58.91     | 39.05       | 71.13          | 48.11            |
> | HealthGPT-M3            | 3.8B       | ✓           | 55.9        | 56.4      | 39.7        | 68.5           | 42.08            |
> | HealthGPT-L14           | 14B        | ✓           | 58.3        | 64.5      | 44.4        | 74.4           | 43.1             |
> | **UniMedVL (Ours)**     | 14B        | ✓           | 61.9        | 75.4      | 53.5        | 85.8*         | 60.75*          |

---

> ### Author Response · Authors · 2025-11-22
> **Response to Weakness 4**
>
> ## Weakness 4.
>
> > 4. Failure case analysis: A detailed investigation of failure cases is necessary to identify the boundary conditions or sample types under which the model underperforms or fails.
>
> A4. We appreciate the reviewer’s request for a more explicit analysis of failure modes and how we handle modality imbalance. Below, we summarize typical failure cases we observed across different tasks. We have added a dedicated subsection, “**A.7 FAILURE CASES AND ANALYSIS**,” to the appendix in our revised manuscript.
> In general, the board categories of failure cases are primarily classified as the following:
>
> ### **Medical Image Generation:**
>  **Global appearance and background artefacts.**
> Although UniMedVL generally produces realistic images across modalities, a characteristic failure mode in text-to-image generation involves embedded text and annotations (**Figure 21**). In some synthesized samples, the model hallucinates spurious on-image text or renders partially legible words, labels, or font styles that do not appear in the corresponding real clinical images or do not match typical acquisition overlays. These artifacts do not alter the main anatomical content but introduce visually unnatural patterns, indicating that handling text-related artifacts remains a challenging aspect of medical image generation. The generated chest X-ray exhibits spurious text overlays (with red boxes highlighting these artifacts), the ultrasound image contains hallucinated text labels that deviate from standard clinical annotations, and the CT scan shows partially corrupted metadata text at the bottom that does not match typical DICOM overlay formatting.
>
> **Structure Preservation in Generation/Editing.**
> For interleaved tasks (e.g., cross-modal synthesis, counterfactual generation), UniMedVL does not always perfectly preserve all spatial structures outside the region being semantically edited (**Figure 22**). For instance, in some counterfactual CXR generations, small devices or lines (e.g., catheters) can become slightly blurred or shifted, even when the main pathological change (e.g., reduction of pleural effusion) is correctly applied. Compared with traditional medical image translation models that operate on strictly registered image pairs, the unified framework can exhibit minor misalignments or shape changes near the edited area, particularly for highly local modifications. **Figure 22** illustrates these challenges across three representative cases: in the brain MRI cross-modal synthesis task (top two rows), the generated images show subtle structural discrepancies in the cerebellar and temporal regions (highlighted by red boxes) compared to the ground truth, and in the chest X-ray counterfactual generation (bottom row), while the model successfully modifies the target pathological region, minor shifts and blurring are observable in the cardiac silhouette boundaries.
>
> ### **Medical Image Understanding:**
> **Clinically demanding quantitative and fine-grained recognition scenarios.**
> We evaluate the performance of UniMedVL on the clinical VQA tasks in GMAI-MMBench. Table 12 summarizes the performance across 18 VQA tasks. In this table, UniMedVL shows relatively low accuracy on temporal recognition tasks involving surgical video clips, including Surgeon Action Recognition (28.70%, **Figure 24**), Surgical Instrument Recognition (30.96%, **Figure 25**), and Surgical Workflow Recognition (47.14%, **Figure 26**). The model also struggles with quantitative assessment and fine-grained recognition tasks, such as Counting (45.75%, **Figure 23**), Image Quality Grading (44.00%, **Figure 27**), and Severity Grading (37.20%, **Figure 28**). This limited performance may arise from the fact that UniMedVL is mainly trained on 2D medical images that lack temporal information, as well as from the limited scale of training data with high-quality fine-grained disease grades or quantitative annotations.
>
> **(Continued below....)**

---

> ### Author Response · Authors · 2025-11-22
> **Continue Response to Weakness 4**
>
> ***Table 12**: UniMedVL performance on GMAI-MMBench validation set. Accuracy across 18 clinical VQA sub-tasks, highlighting challenges in surgical video recognition and quantitative grading.*
>
> | **Task Category**                   | **Accuracy** |
> |-------------------------------------|-------------:|
> | **Overall**                         | **0.607**    |
> | Attribute Recognition               | 0.659        |
> | Blood Vessels Recognition           | 0.593        |
> | Bone                                | 0.623        |
> | Cell Recognition                    | 0.513        |
> | **Counting**                        | **0.457**    |
> | Disease Diagnosis                   | 0.669        |
> | **Image Quality Grading**           | **0.440**    |
> | Microorganism Recognition           | 0.793        |
> | Muscle                              | 0.580        |
> | Nervous Tissue                      | 0.925        |
> | Organ Recognition – Abdomen         | 0.657        |
> | Organ Recognition – Head and Neck   | 0.845        |
> | Organ Recognition – Pelvic          | 0.560        |
> | Organ Recognition – Thorax          | 0.747        |
> | **Severity Grading**                | **0.372**    |
> | **Surgeon Action Recognition**      | **0.287**    |
> | **Surgical Instrument Recognition** | **0.310**    |
> | **Surgical Workflow Recognition**   | **0.471**    |

---

> ### Author Response · Authors · 2025-11-22
> **Response to Weakness 5**
>
> ## Weakness 5.
> >Writing and structure could be improved: The manuscript contains numerous long and compound sentences, making it less readable; a clearer and more concise presentation would improve accessibility.
>
> A5. We appreciate this comment on writing quality and agree that some parts of the manuscript are overly long and dense; we will revise the paper to streamline sentences, improve paragraph structure, and make the presentation clearer and more concise in the revised version.

---

> ### Author Response · Authors · 2025-11-22
> **Response to Questions 1 and 2**
>
> ## Question 1.
>
> > 1. The dataset shows imbalance across modalities (as illustrated in Figure 6). Have the authors experimented with modality-balanced sampling or loss reweighting to mitigate this issue?
>
> A1.  Thank you for raising the issue of modality imbalance. In this work, we do not apply explicit modality-balanced sampling or loss reweighting. First, all understanding tasks are unified under a next-token prediction loss and all image generation tasks under a rectified flow matching loss, so we cannot introduce per-modality loss weights while using this unified objective. Second, balancing modalities via sampling would reduce the number of training samples for modalities with a high data ratio (e.g., Microscopy and MRI, as shown in Figure 7 in the appendix) and could decrease the overall performance of our model. Meanwhile, even modalities with a low data ratio still have substantial absolute sample sizes due to the large overall scale of UniMed-5M. Therefore, UniMedVL achieves the best performance on image generation tasks for modalities with a low data ratio compared to previous image generation models, such as Ultrasound and CFP, **as shown in Table 2 in the manuscript**. To further enhance model performance, a promising future direction is to increase the data scale of modalities with a low data ratio.
>
> ## Question 2.
>
> > 2. In the medical report generation task (Appendix Table 10), UniMedVL underperforms compared to Lingshu-7B. Is this due to differences in task optimization objectives? Could multi-task loss rebalancing improve the model’s performance on this task?
>
> A2.  Thank you for this question. In our setting, medical report generation is trained with the same next-token prediction loss as other models such as Lingshu-7B, so we do not attribute the performance gap to a fundamentally different loss formulation. Rather, it is mainly driven by data scale and allocation: for UniMedVL, we use only about 30K MIMIC-CXR image–report pairs, whereas several baselines exploit the full ≈200K set, and CXR reports account for only a relatively small portion of UniMed-5M because we distribute capacity across multiple imaging modalities and task types.

---

> ### Author Response · Authors · 2025-11-22
> **Response to Questions 3 and 4**
>
> ## Question 3.
>
> > 3. Do the authors plan to release part of the UniMed-5M dataset or model inference weights to facilitate reproducibility and further research within the community?
>
> A3.  Thank you for raising the question of reproducibility. We will release UniMedVL inference weights together with a de-identified subset of UniMed-5M once the remaining license and privacy checks with the source datasets are completed.
> ## Question 4.
>
> > 4. Please also refer to the weaknesses section for additional points that would benefit from clarification or further discussion.
>
> A1. Please refer to the response to the weakness sections.

---

> ### Comment · Reviewer_8GAa · 2025-11-27
>
> I appreciate the detailed clarifications provided in the authors’ response, and I would like to thank the authors for the additional explanations regarding the model design, training strategies, experimental comparisons, and failure case analyses. These replies effectively address my major concerns about the adequacy of the experimental setup and analysis. I also highly recognize the authors’ efforts in constructing the UniMed-5M dataset, as well as their further exploration of a unified understanding-and-generation framework in the medical domain. In terms of data scale, multimodal coverage, and the model’s performance across various tasks, this work indeed carries practical value.
>
> However, I must candidly note that although UniMedVL demonstrates strengths in data contribution and empirical results, its framework-level innovation has not fully convinced me. From the perspective of methodological contribution in a strict sense, I believe the level of innovation remains somewhat conservative.
>
> In summary, I have decided to maintain my original positive rating.

---

> > ### Author Response · Authors · 2025-11-27
> >
> > Dear Reviewer 8GAa,
> >
> > Thank you very much for taking the time to carefully read our rebuttal and for your thoughtful follow-up comments. We are glad that our responses effectively address your major concerns about the adequacy of the experimental setup and analysis, and we sincerely appreciate your recognition of our efforts in constructing the UniMed-5M dataset and exploring a unified understanding-and-generation framework in the medical domain. We are also grateful that you chose to maintain your positive rating.

---

### Official Review · Reviewer_hQhn · 2025-10-30

**Soundness:** 3
**Presentation:** 3
**Contribution:** 2
**Rating:** 4
**Confidence:** 5

**Summary:**

This paper introduces UniMedVL, which is a unified model for medical multimodal understanding and generation. Unlike other models that handle these tasks separately, UniMedVL performs both visual understanding and image generation tasks within a single framework. The model is built around an Observation–Knowledge–Analysis - OKA design that actually reflects how clinicians reason. The paper also releases a large-scale dataset UniMed-5M, which reformats diverse medical datasets into unified multimodal pairs across eight imaging modalities. The experiments on benchmarks such as VQA-RAD, SLAKE, and OmniMedVQA show competitive results compared to specialized models.

**Strengths:**

- **Clinically relevant motivation**: The proposed Observation–Knowledge–Analysis (OKA) framework reflects the natural workflow of medical diagnosis—observe findings, apply knowledge, and form conclusions.
- **Large and diverse dataset**: The introduction of UniMed-5M, covering multiple imaging modalities and medical domains, represents a valuable resource for the community.
- **Clear writing**: The paper is well written and clearly structured. Figures and tables illustrate key concepts, and the experimental findings are communicated in a logical way.

**Weaknesses:**

- **Limited theoretical insights**: The paper provides little explanation of why the Observation–Knowledge–Analysis framework leads to better performance. The benefits appear mostly empirical, without any deeper analysis of internal representations or cross-modal information flow.
- **High model complexity/computational costs**: The dual-encoder architecture combined with a Mixture-of-Transformers (MoT) module increases computational costs, especially compared to unified models which use one shared multimodal transformer (e.g., BLIP3o). Also, the paper does not report training/inference speed or trade-off comparisons to such unified models.
- **Overlaps with HealthGPT**: The paper shows strong similarities to HealthGPT, which also aims to unify medical understanding and generation within a single model. The novelty compared to HealthGPT—particularly in architecture: MoE vs MoT - is not clearly established.

**Questions:**

1. In Table 1, the baseline GMAI-VL (7B) outperforms or performs comparably to the proposed UniMedVL (14B) on visual understanding benchmarks, despite using half as many parameters. Could the authors elaborate on this observation? What factors might explain why the larger unified model does not surpass the smaller baseline in understanding tasks?
2. The proposed UniMedVL uses MoT design for explicit expert routing for understanding and generation. How does this compare to unified multimodal models like BLIP-3o, which uses one shared multimodal transformer? A brief discussion of these design trade-offs would clarify the novelty and advantages of the MoT approach.
3. How does the proposed model UniMedVL differ from HealthGPT, which also unifies medical understanding and generation? Could the authors clarify the unique contributions beyond those in the HealthGPT paper?
4. The MoT design likely increases computational cost and memory usage. Could the authors provide quantitative estimates of the additional training time and inference overhead compared to a single-expert baseline?
5. Table 10 shows that UniMedVL performs worse on the MIMIC-CXR report generation benchmark compared to some baselines. Could the authors clarify potential reasons for this drop?

---

> ### Author Response · Authors · 2025-11-22
> **Response to Weakness 1**
>
> ## Weakness 1.
>
> > - Limited theoretical insights: The paper provides little explanation of why the Observation–Knowledge–Analysis framework leads to better performance. The benefits appear mostly empirical, without any deeper analysis of internal representations or cross-modal information flow.
>
> A: Thank you for this valuable suggestion. We agree that a theoretical analysis of the Observation–Knowledge–Analysis (OKA) framework could improve the clarity and presentation of our submission. While a comprehensive theoretical analysis is beyond the scope of this work, we offer the following explanation that combines theoretical intuitions with empirical observations.
>
> The OKA framework is theoretically motivated by the hierarchical nature of clinical reasoning and the principles of curriculum learning in representation theory. Rather than being an arbitrary three-stage pipeline, OKA systematically addresses three fundamental bottlenecks in medical multimodal learning:
>
> **Observation level.** At the Observation level, we collect and reformat large-scale medical multimodal datasets from image understanding, generation, and interleaved tasks into uniform multimodal input–output pairs, introducing UniMed-5M as the initial building block for unifying diverse medical tasks. This data-level unification establishes the upper bound for model performance [1].
>
> **Knowledge level.** At the Knowledge level, we introduce a Progressive Curriculum Learning strategy that includes Foundation Training, Instruction Tuning, and Unified Multimodal Training:
>
>    1. **Foundation Training stage:** We utilize the entire UniMed-5M to establish basic medical pattern recognition by adapting the model to medical data distributions.
>
>    2. **Instruction Tuning stage:** We further fine-tune the stage-1 model using high-quality instruction data to improve its instruction-following capabilities for solving diverse medical tasks and strengthen medical task-specific guidance [2].
>
>    3. **Unified Multimodal Training stage:** Continuing from the stage-2 model, we further enhance the bidirectional understanding–generation capabilities of UniMedVL using our formulated interleaved tasks.
>
> **Analysis level.** UniMedVL embodies the principle that medical diagnosis is inherently a multi-input, multi-output process, and adopts a MoT design that maintains two Transformer experts for medical understanding and generation tasks respectively. This design addresses the task interference problem [3] while preserving beneficial cross-task transfer. This approach is validated by our ablation studies, which show mutual performance improvements between understanding and generation.
>
> Consequently, UniMedVL achieves state-of-the-art overall performance on image understanding, generation, and interleaved tasks compared with previous unified models such as HealthGPT.
>
> [1] Tripuraneni N, Jordan M, Jin C. On the theory of transfer learning: The importance of task diversity. NIPS 2020.
> [2] Liu H, Li C, Wu Q, et al. Visual instruction tuning. NIPS 2023.
> [3] Wu C, Chen X, Wu Z, et al. Janus: Decoupling visual encoding for unified multimodal understanding and generation. CVPR 2025.

---

> ### Author Response · Authors · 2025-11-22
> **Response to Weakness 2**
>
> ## Weakness 2.
>
> > - High model complexity/computational costs: The dual-encoder architecture combined with a Mixture-of-Transformers (MoT) module increases computational costs, especially compared to unified models which use one shared multimodal transformer (e.g., BLIP3o). Also, the paper does not report training/inference speed or trade-off comparisons to such unified models.
>
> A2. Thanks for your comments. During inference, UniMedVL activates only a single Transformer expert, so the effective path is single-branch despite the dual-track MoT design. To directly address the reviewer’s concern about latency and deployment efficiency, we have added an explicit efficiency analysis **in Appendix A.8 (Table 13. Efficiency Evaluation)**, comparing UniMedVL with other unified medical multimodal models under batch size 1.
>
> For text-to-image generation, UniMedVL (14B) requires 40.03 TFLOPs per image and 28.39 GB peak memory, whereas BLIP3-o 8B requires 142.59 TFLOPs per image and 25.55 GB. This corresponds to ≈3.6× lower compute per image for UniMedVL with only ~11% higher peak memory (28.39 vs. 25.55 GB), even though UniMedVL uses a dual-encoder and MoT design and has almost twice the parameters. UniMedVL’s compute cost is also close to that of Janus 7.42B (35.56 TFLOPs/image) while providing a larger unified model.
>
> For VQA on GMAI-MMBench, UniMedVL achieves 25.86 tokens/s with 2.256 TFLOPs per sample and 28.25 GB peak memory, compared with BLIP3-o’s 30.40 tokens/s, 9.307 TFLOPs per sample, and 18.21 GB. Thus, UniMedVL attains a ≈4.1× reduction in FLOPs per sample with comparable throughput (about 85% of BLIP3-o’s tokens/s), at the cost of higher peak memory due to the dual-encoder design. Compared to another 14B unified model, HealthGPT-L14 (12.57 tokens/s, 3.009 TFLOPs, 29.22 GB), UniMedVL is roughly twice as fast in throughput and more compute-efficient.
>
> Overall, the measurements show that, despite its dual-track MoT architecture, UniMedVL maintains competitive latency and compute efficiency relative to single-path baselines, and the main overhead manifests as a moderate increase in peak memory rather than prohibitive inference cost.
>
>
> *Table 13: **Efficiency Evaluation.** Comparison of throughput and computational costs across unified medical multimodal models with batch size 1.*
>
> **(a) Image Generation Throughput**
>
> *Warm up with 10 images and measure efficiency over 20 images*
>
> | Model       | Parameters | FLOPs/Image (TFLOPs) | Peak Mem (GB) |
> |------------|------------|----------------------|---------------|
> | Janus      | 1B         | 10.01                | 5.19          |
> | HealthGPT-M3 | 3.8B     | 15.22                | 10.23         |
> | Janus      | 7.42B      | 35.56                | 17.10         |
> | BLIP3-o    | 8B         | 142.59               | 25.55         |
> | **UniMedVL** | **14B**  | **40.03**            | **28.39**     |
>
> **(b) VQA Understanding Throughput (GMAI-MMBench validation set)**
>
> *Warm up with 50 questions and measure efficiency over 150 VQA questions*
>
> | Model         | Parameters | Tokens/s | FLOPs/Sample (TFLOPs) | Peak Mem (GB) |
> |--------------|------------|----------|------------------------|---------------|
> | Janus        | 1B         | 70.11    | 0.498                  | 4.46          |
> | HealthGPT-M3 | 3.8B       | 22.13    | 1.304                  | 8.79          |
> | Janus        | 7.42B      | 52.94    | 1.894                  | 14.59         |
> | BLIP3-o      | 8B         | 30.40    | 9.307                  | 18.21         |
> | **UniMedVL** | **14B**    | **25.86**| **2.256**              | **28.25**     |
> | HealthGPT-L14| 14B        | 12.57    | 3.009                  | 29.22         |

---

> ### Author Response · Authors · 2025-11-22
> **Response to Weakness 3**
>
> ## Weakness 3.
>
> > - Overlaps with HealthGPT: The paper shows strong similarities to HealthGPT, which also aims to unify medical understanding and generation within a single model. The novelty compared to HealthGPT—particularly in architecture: MoE vs MoT - is not clearly established.
>
> A3. Thank you for the opportunity to clarify the novelty beyond a superficial MoE–vs–MoT comparison with HealthGPT. Conceptually, UniMedVL differs from HealthGPT along three axes:
>
> (i) **Data and scope.** UniMed-5M is substantially larger than the dataset used by HealthGPT, which contains only 700K samples. UniMed-5M is explicitly constructed to cover understanding, generation, and interleaved tasks, whereas HealthGPT is trained only on separate understanding and generation task data without interleaved task integration.
>
> (ii) **Task design.** We explicitly define a broad family of interleaved tasks—such as counterfactual image generation and virtual staining—where the model needs to understand the input images and generate corresponding images according to the instructions. Furthermore, these tasks constitute a dedicated third stage of unified multimodal training in our proposed Progressive Curriculum Training. HealthGPT does not introduce such interleaved task formulations or a corresponding training stage, and therefore cannot directly support these clinically important interleaved tasks within a single unified setup.
>
> (iii) **Architecture and representation learning.** Instead of using task-specific H-LoRA adapters, which still require separate checkpoints for understanding and generation tasks in HealthGPT, UniMedVL adopts a MoT design where tokens from understanding and generation tasks are processed in shared self-attention layers, enabling genuine joint representations and a single unified checkpoint that handles all task types. This design yields consistently stronger performance than HealthGPT on both understanding and generation tasks.
>
> UniMedVL provides stronger unification than HealthGPT through larger data, interleaved-task training, and a single MoT-based backbone design.

---

> ### Author Response · Authors · 2025-11-22
> **Response to Question 1**
>
> ## Question 1.
>
> > 1. In Table 1, the baseline GMAI-VL (7B) outperforms or performs comparably to the proposed UniMedVL (14B) on visual understanding benchmarks, despite using half as many parameters. Could the authors elaborate on this observation? What factors might explain why the larger unified model does not surpass the smaller baseline in understanding tasks?
>
> A4. Thank you for highlighting the comparison with GMAI-VL. We would like to clarify that, although UniMedVL is a 14B-parameter model, it only activates 7B parameters when performing understanding tasks, so the inference capacity is comparable to GMAI-VL-7B rather than strictly larger.
>
> In **Table 1 in the manuscript**, UniMedVL surpasses GMAI-VL on SLAKE and PathVQA, remains slightly weaker on VQA-RAD, OmniMedVQA, and GMAI-MMBench yet still outperforms other understanding and unified models, and achieves a higher average score across these understanding benchmarks (67.47 for UniMedVL vs. 65.85 for GMAI-VL).
>
> We also note that UniMedVL is trained with less understanding data (≈4.0M vs. GMAI-VL’s 5.5M), **as shown in Table 8 in the Appendix**, since part of UniMed-5M is reserved for generation and interleaved tasks; thus, UniMedVL attains better aggregate understanding performance with the same number of activated parameters and less understanding training data compared with GMAI-VL.

---

> ### Author Response · Authors · 2025-11-22
> **Response to Question 2**
>
> ## Question 2.
>
> > 2. The proposed UniMedVL uses MoT design for explicit expert routing for understanding and generation. How does this compare to unified multimodal models like BLIP-3o, which uses one shared multimodal transformer? A brief discussion of these design trade-offs would clarify the novelty and advantages of the MoT approach.
>
> **A2.** We thank the reviewer for this insightful question regarding the design trade-offs between our MoT-based architecture and unified multimodal transformers such as BLIP-3o. UniMedVL, which utilizes the MoT design with shared self-attention layers, requires more GPU memory and a longer training time during training compared with models that use a single shared multimodal transformer, such as BLIP-3o. However, during inference, UniMedVL activates only half of its total parameters, achieving a similar inference time to BLIP-3o with only a slight increase in GPU memory usage. **Table 5(a–b)** quantifies these efficiency trade-offs.
>
> It is also worth noting that BLIP-3o is primarily trained on general-domain data and has limited exposure to large-scale medical imaging datasets. To provide a fairer comparison with unified multimodal models that are explicitly tailored to the medical domain, we compare UniMedVL with **HealthGPT-M3** and **HealthGPT-L14**, both of which adopt a **single multimodal transformer backbone with H-LoRA** for medical understanding and generation. As shown in **Table R1**, UniMedVL attains higher **Performance-vs-FLOPs** score compared to HealthGPT, indicating that the MoT-based design provides a more favorable trade-off between computational cost and task performance.
> The Performance-vs-FLOPs is defined as the average accuracy on five medical understanding benchmarks divided by the logarithm of FLOPs per sample, i.e., $\text{Perf-vs-FLOPs} = \frac{\text{Acc}}{\log_{10}(\text{FLOPs})}$.
>
> Moreover, prior work has shown that task interference can occur between understanding and generation tasks [1]. In contrast, BLIP-3o uses a single shared multimodal transformer, which is architecturally simple but can make it difficult to achieve high performance on both medical image understanding and generation tasks simultaneously due to task interference. In UniMedVL, the MoT design maintains two transformer experts for understanding and generation tasks, respectively, while sharing self-attention layers between them. This mitigates such interference and encourages learning joint representations across understanding and generation tasks.
>
>
> **Table 5: Efficiency Evaluation.** Comparison of throughput and computational costs across unified medical multimodal models with batch size 1.
>
> **(a) Image Generation Throughput**
>
> | Model       | Architecture        | Parameters | FLOPs/Image (TFLOPs) | Peak Mem (GB) | Relative FLOPs vs BLIP3-o |
> |-------------|----------------------|------------|------------------------|----------------|----------------------------|
> | BLIP3-o     | Single Transformer   | 8B         | 142.59                 | 25.55          | 1×          |
> | **UniMedVL**| **MoT**              | **14B**    | **40.03**              | **28.39**      | **~3.6×** fewer   |
>
>
> **(b) VQA Understanding Inference Throughput (GMAI-MMBench validation set)**
>
> | Model       | Architecture        | Parameters | Tokens/s | FLOPs/Sample (TFLOPs) | Peak Mem (GB) | Relative FLOPs vs BLIP3-o |
> |-------------|----------------------|------------|----------|------------------------|----------------|----------------------------|
> | BLIP3-o     | Single Transformer   | 8B         | 30.40    | 9.307                  | 18.21          | 1×        |
> | **UniMedVL**| **MoT**              | **14B**    | **25.86**| **2.256**              | **28.25**      | **~4.1x** fewer   |
>
>
>
> **Table R1** Comparison with medical-domain unified models.
>
> | Model             | Architecture                      | Params | Tokens/s | FLOPs/Sample (TFLOPs) | Average Performance (%) | Perf-vs-FLOPs|
> |-------------------|-----------------------------------|--------|----------|------------------------|-------------------------------------|----------------|
> | HealthGPT-M3      | Single Transformer + H-LoRA       | 3.8B   | 22.13    | 1.304                  | 52.52                               | 4.33           |
> | HealthGPT-L14     | Single Transformer + H-LoRA       | 14B    | 12.57    | 3.009                  | 56.94                               | 4.56           |
> | **UniMedVL (Ours)** | **MoT (2 experts)**             | **14B**| **25.86**| **2.256**              | **67.47**                           | **5.46**       |
>
>
> [1] Wu C, Chen X, Wu Z, et al. Janus: Decoupling visual encoding for unified multimodal understanding and generation. CVPR'2025.

---

> ### Author Response · Authors · 2025-11-22
> **Response to Questions 3 and 4**
>
> ## Question 3.
>
> > How does the proposed model UniMedVL differ from HealthGPT, which also unifies medical understanding and generation? Could the authors clarify the unique contributions beyond those in the HealthGPT paper?
>
> A3. Please refer to the response to the Weaknesses 3.
> ## Question 4.
>
> > The MoT design likely increases computational cost and memory usage. Could the authors provide quantitative estimates of the additional training time and inference overhead compared to a single-expert baseline?
>
> A4. Thank you for raising the question on the computational and memory overhead of the MoT design. We establish single-expert baselines by deactivating one branch in the MOT architecture to control experimental settings. We evaluate both training time and computational overhead by measuring parameters and seconds per iteration under fixed token input length. For each configuration, we train models for 1000 iterations from first-stage checkpoints with the specified data configuration. We then extrapolate total training time for single-expert baselines via the ratio of 1000 iterations to total iterations. All benchmarks use 8 A800 GPUs.
>
> Table R1 . Training efficiency comparison between UniMedVL (MoT) and single-expert baselines constructed by deactivating one branch.
>
> | Configuration       | Active Params (B) | Total Params (B) | sec per iteration | Relative Speed | GPU Hours 8xA800 | Training Type  |
> |---------------------|----------------:|---------------:|------------------:|---------------:|-----------------:|----------------|
> | Understanding-only  |           7.63  |          14.61 |             3.195 |        1.156×  |             1952 | Single-expert  |
> | Generation-only     |           7.63  |          14.61 |             3.232 |        1.143×  |             1952 | Single-expert  |
> | UniMedVL (MoT)      |          14.58  |          14.61 |             3.695 |          1.0×  |             2258 | Joint (MoT)    |
>
> The results indicate that our MoT architecture increase training time of 15.6% from 1952 A800 GPU hours to 2258 GPU hours and double the training parameters compared to a single-expert baseline.
>
> Regarding the inference overhead, we have compared the inference cost of UniMedVL with BLIP-3o-8B in our response to Weakness 2, as BLIP-3o uses a single shared multimodal transformer. UniMedVL does not introduce a significant inference burden compared to BLIP-3o.

---

> ### Author Response · Authors · 2025-11-22
> **Response to Question 5**
>
> ## Question 5.
>
> > Table 10 shows that UniMedVL performs worse on the MIMIC-CXR report generation benchmark compared to some baselines. Could the authors clarify potential reasons for this drop?
>
> A5.  We appreciate the reviewer’s attention to the performance gap on the MIMIC-CXR report generation benchmark. We believe this gap is mainly due to the data scale of MIMIC-CXR. In our training, we used only about 30K MIMIC-CXR image–report pairs (whereas several baselines leverage the full ≈200K set), and CXR reports constitute only a relatively small fraction of UniMed-5M because we deliberately allocate capacity across multiple modalities and interleaved tasks.

---

### Official Review · Reviewer_nhqo · 2025-11-01

**Soundness:** 3
**Presentation:** 3
**Contribution:** 3
**Rating:** 6
**Confidence:** 4

**Summary:**

This paper tackles the divide between medical image understanding and generation by introducing the Observation–Knowledge–Analysis (OKA) paradigm aligned with clinical workflows. Observation: UniMed-5M (>5.6M samples) reformats heterogeneous unimodal sources into paired multimodal inputs to support unified learning. Knowledge: Progressive Curriculum Learning builds capability in three stages (foundation pretraining, instruction tuning, unified multimodal training). Analysis: UniMedVL unifies understanding and generation within one architecture using dual visual modules (EViT for encoding, EVAE for generation) and a Mixture-of-Transformer-Experts, trained with a combined objective.

Empirically, UniMedVL reports strong results on five medical image understanding benchmarks and generation quality comparable to specialized models across eight imaging modalities. The authors emphasize that a unified architecture enables bidirectional knowledge sharing, where generative training can enhance visual understanding features. Overall, the work positions unified modeling and curriculum design as a path to narrow the understanding–generation gap in medical AI.

**Strengths:**

1. The OKA framing maps clinical reasoning onto model design and training, offering genuine conceptual novelty beyond task-specific pipelines. A comprehensive experimental suite spanning understanding, generation, and interleaved tasks across multiple modalities and datasets, evaluated with diverse metrics, strengthens the empirical case for generality.

2. Large, diverse UniMed-5M (>5.6M samples, nine modalities) enables multimodal learning at scale. Its breadth across modalities and tasks, plus the reformulation of unimodal sources into paired multimodal examples, mitigates data scarcity and supports both understanding and generation pretraining.

3. Coherent training and architecture (Progressive Curriculum Learning; EViT + EVAE + MoT) with evidence of bidirectional knowledge sharing. The staged curriculum builds capability from base pretraining to unified multimodal learning, while the dual visual modules and MoT allow seamless task switching without checkpoint reloads and show transfer where generation benefits understanding.

**Weaknesses:**

Despite the aforementioned strengths, each has potential limitations, and in some cases the supporting evidence is less robust than the authors claim.

1.	What “unified” really means
- I acknowledge the practical need to separate objectives across understanding and generation, but I remain concerned. Despite the unified claim, distinct extractors (EViT for understanding, EVAE for generation) and partitioned MoT experts indicate coordinated specialization rather than a deeply shared joint representation. The integration feels coarse, leaving its “unified” status uncertain, and the dual-track design likely raises inference cost versus single-path baselines, impacting latency and deployment efficiency.

2.	Dataset construction and quality control
- A substantial portion of UniMed-5M is synthesized via LLM captioning, which introduces risks of hallucination and bias, especially for counterfactuals. Expert review on about 5% may not capture rare or clinically nuanced cases. Templateization and VLLM captioning can standardize style while injecting model or template biases, reducing real-world linguistic diversity.

3.	Limits of the VAE evaluation
- The EVAE is fixed from a general-purpose, non-medical pretraining. Medical images contain subtle anatomy and pathology that may be underrepresented in generic VAEs. Reconstruction metrics (rFID, PSNR, SSIM) do not directly reflect clinical fidelity for small but important lesions, and specialized medical VAEs outperform in some modalities.

4.	Lack of details in methodological robustness and generalization
- Progressive Curriculum Learning bundles multiple factors (sampling, learning rates, ViT training schedule) without isolating causal effects. The loss weight $\alpha$=4 appears empirically chosen, with no sensitivity study. These gaps limit confidence in generalization beyond the reported settings.

**Questions:**

n/a

---

> ### Author Response · Authors · 2025-11-22
> **Response to Weakness 1**
>
> ## Weakness 1.
>
> > 1. What “unified” really means
> > - I acknowledge the practical need to separate objectives across understanding and generation, but I remain concerned. Despite the unified claim, distinct extractors (EViT for understanding, EVAE for generation) and partitioned MoT experts indicate coordinated specialization rather than a deeply shared joint representation. The integration feels coarse, leaving its “unified” status uncertain, and the dual-track design likely raises inference cost versus single-path baselines, impacting latency and deployment efficiency.
>
> A1. Thank you for raising this concern about what we mean by “unified.” In our work, “unified” means that the proposed UniMedVL can simultaneously accomplish medical image understanding and generation tasks. To achieve this, UniMedVL leverages EViT and EVAE to encode the inputs of both understanding and generation tasks into a single unified token sequence. This token sequence is then processed by the Mixture-of-Transformer (MoT) experts with shared self-attention layers, which learn the contextual relationships within the sequence. Meanwhile, we design a Progressive Curriculum Learning strategy to jointly optimize the understanding and generation objectives using UniMed-5M with uniform multimodal input–output pairs. With this model architecture and training paradigm, UniMedVL learns a joint representation for both understanding and generation tasks.
>
> During inference, UniMedVL activates only a single Transformer expert, so the effective path is single-branch despite the dual-track MoT design. To directly address the reviewer’s concern about latency and deployment efficiency, we have added an explicit efficiency analysis in **Appendix A.8 (Table 13. Efficiency Evaluation)**, comparing UniMedVL with other unified medical multimodal models under batch size 1.
>
> **For text-to-image generation**, UniMedVL (14B) requires 40.03 TFLOPs per image and 28.39 GB peak memory, whereas BLIP3-o 8B requires 142.59 TFLOPs per image and 25.55 GB. This corresponds to ≈3.6× lower compute per image for UniMedVL with only ~11% higher peak memory (28.39 vs. 25.55 GB), even though UniMedVL uses a dual-encoder and MoT design and has almost twice the parameters. UniMedVL’s compute cost is also close to that of Janus 7.42B (35.56 TFLOPs/image) while providing a larger unified model.
>
> **For VQA on GMAI-MMBench**, UniMedVL achieves 25.86 tokens/s with 2.256 TFLOPs per sample and 28.25 GB peak memory, compared with BLIP3-o’s 30.40 tokens/s, 9.307 TFLOPs per sample, and 18.21 GB. Thus, UniMedVL attains a ≈4.1× reduction in FLOPs per sample with comparable throughput (about 85% of BLIP3-o’s tokens/s), at the cost of higher peak memory due to the dual-encoder design. Compared to another 14B unified model, HealthGPT-L14 (12.57 tokens/s, 3.009 TFLOPs, 29.22 GB), UniMedVL is roughly twice as fast in throughput and more compute-efficient.
>
> Overall, the measurements in **Appendix A.8, Table 13** show that, despite its dual-track MoT architecture, UniMedVL maintains competitive latency and compute efficiency relative to single-path baselines, and the main overhead manifests as a moderate increase in peak memory rather than prohibitive inference cost.
>
> ***Table 13** : **Efficiency Evaluation.** Comparison of throughput and computational costs across unified medical multimodal models with batch size 1.*
>
> **(a) Image Generation Throughput**
>
> *Warm up with 10 images and measure efficiency over 20 images*
>
> | Model       | Parameters | FLOPs/Image (TFLOPs) | Peak Mem (GB) |
> |------------|------------|----------------------|---------------|
> | Janus      | 1B         | 10.01                | 5.19          |
> | HealthGPT-M3 | 3.8B     | 15.22                | 10.23         |
> | Janus      | 7.42B      | 35.56                | 17.10         |
> | BLIP3-o    | 8B         | 142.59               | 25.55         |
> | **UniMedVL** | **14B**  | **40.03**            | **28.39**     |
>
> **(b) VQA Understanding Throughput (GMAI-MMBench validation set)**
>
> *Warm up with 50 questions and measure efficiency over 150 VQA questions*
>
> | Model         | Parameters | Tokens/s | FLOPs/Sample (TFLOPs) | Peak Mem (GB) |
> |--------------|------------|----------|------------------------|---------------|
> | Janus        | 1B         | 70.11    | 0.498                  | 4.46          |
> | HealthGPT-M3 | 3.8B       | 22.13    | 1.304                  | 8.79          |
> | Janus        | 7.42B      | 52.94    | 1.894                  | 14.59         |
> | BLIP3-o      | 8B         | 30.40    | 9.307                  | 18.21         |
> | **UniMedVL** | **14B**    | **25.86**| **2.256**              | **28.25**     |
> | HealthGPT-L14| 14B        | 12.57    | 3.009                  | 29.22         |

---

> ### Author Response · Authors · 2025-11-22
> **Response to Weakness 2**
>
> ## Weakness 2.
>
> > 2. Dataset construction and quality control
> >
> > * A substantial portion of UniMed-5M is synthesized via LLM captioning, which introduces risks of hallucination and bias, especially for counterfactuals.
> > * Expert review on about 5% may not capture rare or clinically nuanced cases.
> > * Templateization and VLLM captioning can standardize style while injecting model or template biases, reducing real-world linguistic diversity.
>
> A2. Thank you for the thoughtful observations on aspects of data synthesis, expert validation scope, and stylistic consistency within UniMed-5M.
>
> ### Data Provenance
>
> UniMed-5M is constructed from human-validated open-source datasets, including expert-annotated imaging datasets, peer-reviewed articles, and authentic radiology reports. We reorganize these datasets into a unified format without altering original information. Although construction pipelines of some collected datasets leverage large language models, LLM outputs are grounded in original validated materials, with each instance maintaining traceability to its clinical source and preserving factual content of original medical findings.
>
> ### Expert Review
>
> As illustrated in Figure 7 in the appendix, to ensure comprehensive expert review coverage across a broad spectrum of clinical cases, we classify UniMed-5M samples by image modality and anatomical structure. Subsequently, we sample 5% of training data for each medical imaging modality and anatomical structure, upon which we conduct broad-based expert review through surveys and questionnaires.
>
> ### Bias Mitigation
>
> Regarding the templateization issue, we acknowledge its existence and recognize the inherent trade-off between medical captioning diversity and adherence to clinical standards. To mitigate this concern, we use GPT-4o to construct a comprehensive library of instructions for each medical task. Subsequently, we proofread the candidate instructions in collaboration with medical experts to identify and mitigate potential biases, thereby ensuring alignment with established practices in clinical settings [1–3]. Through this approach, we aim to enhance the real-world linguistic diversity of UniMed-5M while also mitigating template-induced or model-induced biases to preserve clinical authenticity.
>
> [1] Li C, Wong C, Zhang S, et al. LLaVA-Med: Training a Large Language-and-Vision Assistant for the Biomedical Domain. NeurIPS ’23.
>
> [2] Singhal K, Azizi S, Tu T, et al. Large Language Models Encode Clinical Knowledge. Nature 2023.
>
> [3] Chen J, Gui C, Ouyang R, et al. HuatuoGPT-Vision: Injecting Medical Visual Knowledge into Multimodal LLMs at Scale. EMNLP 2024.

---

> ### Author Response · Authors · 2025-11-22
> **Response to Weakness 3**
>
> ## Weakness 3.
>
> > 3. Limits of the VAE evaluation
> > - The EVAE is fixed from a general-purpose, non-medical pretraining. Medical images contain subtle anatomy and pathology that may be underrepresented in generic VAEs. Reconstruction metrics (rFID, PSNR, SSIM) do not directly reflect clinical fidelity for small but important lesions, and specialized medical VAEs outperform in some modalities.
>
>
> A3. Thank you for pointing out this concern about using a frozen, non-medical VAE for images with small but clinically important lesions. Following this suggestion, we have added comprehensive VAE evaluation results in **Appendix A.1 Figure 6**. Figure 6 presents side-by-side comparisons of original medical images containing small lesions and their reconstructions from the frozen general-purpose VAE. Specifically, we evaluate three representative cases: an endoscopy image containing a polyp, a dermoscopy image with a skin lesion exhibiting globules, and an X-ray image showing a fracture. The reconstructed images exhibit visual features highly similar to the originals, demonstrating that the frozen VAE faithfully preserves small but clinically important lesions while maintaining high reconstruction fidelity.

---

> ### Author Response · Authors · 2025-11-22
> **Response to Weakness 4**
>
> ## Weakness 4.
>
> > 1. Lack of details in methodological robustness and generalization
> > - Progressive Curriculum Learning bundles multiple factors (sampling, learning rates, ViT training schedule) without isolating causal effects. The loss weight = 4 appears empirically chosen, with no sensitivity study. These gaps limit confidence in generalization beyond the reported settings.
>
> **A4.** We appreciate the concern regarding curriculum and loss design justification. In the manscript,  **Table 5** ablates the principal components of Progressive Curriculum Learning that affect model performance. Specifically, training ViT on the understanding task yields substantial gains in understanding while preserving generation performance (One-Stage-Joint-Base vs. H-Joint-Base in Stage 1).
>
> Regarding data sampling ratios and loss weights, we examine several extreme configurations. The results show that combining understanding and generation data and using both understanding and generation loss functions improves overall performance relative to single-task training (H-Joint-Base vs. C-G-only and B-U-only in Stage 1).
>
> Concerning learning rates, the general principle follows that of traditional optimization, with the critical distinction that we prioritize training stability, as each run with 14B parameters demands substantial computational resources. To determine a suitable learning rate, we conduct a preliminary study examining three candidates: $1 \times 10^{-6}$, $1 \times 10^{-5}$, and $1 \times 10^{-4}$. We train the models on the first-stage training dataset for 500 iterations and monitor both $L_{\mathrm{NTP}}$ and $L_{\mathrm{flow}}$. The selection principle satisfies two objectives: (i) an adequate rate of loss decrease and (ii) training stability. Large learning rates (e.g., $1 \times 10^{-4}$) induce instability, while excessively small learning rates (e.g., $1 \times 10^{-6}$) are insufficient for efficiently incorporating medical knowledge. The results are shown in the following tables.
>
> **$L_{\mathrm{NTP}}$ loss:**
>
> | Learning Rate | Step 0 | Step 100 | Step 200 | Step 300 | Step 400 | Step 500 |
> |--------------:|-------:|---------:|---------:|---------:|---------:|---------:|
> | 1.00E-06      | 12.75  | 12.61    | 12.45    | 12.26    | 12.11    | 11.89    |
> | 1.00E-05      | 12.75  | 11.84    | 10.31    | 9.568    | 9.046    | 8.271    |
> | 1.00E-04      | 12.75  | 8.623    | 7.576    | 6.626    | 6.628    | 6.076    |
>
> **$L_{\mathrm{flow}}$ loss:**
>
> | Learning Rate | Step 0 | Step 100 | Step 200 | Step 300 | Step 400 | Step 500 |
> |--------------:|-------:|---------:|---------:|---------:|---------:|---------:|
> | 1.00E-06      | 2.857  | 2.454    | 2.191    | 1.887    | 1.716    | 1.653    |
> | 1.00E-05      | 2.857  | 1.656    | 1.417    | 1.380    | 1.357    | 1.287    |
> | 1.00E-04      | 2.857  | 1.635    | 1.411    | 1.350    | 1.300    | 0.8346   |
>
> Based on empirical observations in the two tables, we analyze loss trajectories under the three learning rates. The rate $1 \times 10^{-6}$ exhibits slow but monotonic descent for both $L_{\mathrm{NTP}}$ and $L_{\mathrm{flow}}$. The rate $1 \times 10^{-4}$ demonstrates rapid initial convergence but introduces training instability. In contrast, $1 \times 10^{-5}$ achieves stable and consistent loss reduction for $L_{\mathrm{NTP}}$ while maintaining moderate $L_{\mathrm{flow}}$ loss descent.
>
> Comparing $1 \times 10^{-4}$ and $1 \times 10^{-5}$ further justifies our selection. Although $1 \times 10^{-4}$ yields faster loss reduction, our multi-stage training strategy obviates the need for rapid descent of loss values. Therefore, we prioritize a conservative learning rate that guarantees stability and the rate $1 \times 10^{-5}$ satisfies this criterion.

---

### Official Review · Reviewer_y8Tr · 2025-11-02

**Soundness:** 3
**Presentation:** 3
**Contribution:** 2
**Rating:** 6
**Confidence:** 4

**Summary:**

This paper proposes UniMedVL, a unified framework designed to perform both medical image understanding (e.g., Visual Question Answering, report generation) and medical image generation (e.g., text-to-image synthesis, cross-modal translation) within a single model architecture. The work is structured around a novel Observation-Knowledge-Analysis (OKA) paradigm, which mirrors the clinical diagnostic process:

Observation (Data-level): The authors construct UniMed-5M, a large-scale dataset of over 5.6 million multimodal medical samples, reformatted from various public sources to create uniform input-output pairs.

Knowledge (Feature-level): They introduce Progressive Curriculum Learning, a three-stage training strategy (Foundation Training, Instruction Tuning, Unified Multimodal Training) to systematically build the model's capabilities from basic pattern recognition to sophisticated multimodal reasoning.

Analysis (Task-level): The core contribution is the UniMedVL model, which uses a dual-encoder (ViT for understanding, VAE for generation) and mixture-of-experts transformer architecture to handle diverse tasks without switching checkpoints.

Extensive experiments demonstrate that UniMedVL achieves state-of-the-art or highly competitive performance on five medical image understanding benchmarks while matching specialized models in generation quality across eight imaging modalities. A key finding is the existence of bidirectional knowledge transfer, where joint training on understanding and generation tasks mutually enhances performance in both domains.

**Strengths:**

The paper introduces a novel, clinically-inspired OKA framework and is the first to demonstrate a truly unified model for medical multimodal understanding and generation within a single, end-to-end architecture, outperforming models that require task-specific components.

The massive scale and careful construction of the UniMed-5M dataset will be helpfule for the whole community (if publicly accessiable).

This work represents a substantial leap towards integrated clinical AI assistants. By unifying capabilities, it addresses a critical gap in the field. The demonstrated positive synergy between understanding and generation tasks challenges conventional wisdom and opens up new research avenues.

**Weaknesses:**

While the expert validation of individual outputs is excellent, a deeper discussion on the model's performance in end-to-end clinical decision-support scenarios is missing. For instance, how does the model's generated image and its explanation directly influence a simulated diagnostic or treatment planning task compared to a human expert or a pipeline of specialized models?

**Questions:**

Ablation on VAE Fine-tuning: The paper justifies using a frozen, general-purpose VAE with strong reconstruction metrics. However, was any experimentation done with a lightly fine-tuned medical VAE? A small ablation could clarify if there are diminishing returns or potential risks of domain-specific adaptation that were avoided.

---

> ### Author Response · Authors · 2025-11-22
> **Response to Weakness 1**
>
> ## Weakness 1.
>
> > While the expert validation of individual outputs is excellent, a deeper discussion on the model's performance in end-to-end clinical decision-support scenarios is missing. For instance, how does the model's generated image and its explanation directly influence a simulated diagnostic or treatment planning task compared to a human expert or a pipeline of specialized models?
>
> A1: We appreciate the reviewer's concern regarding end-to-end clinical decision-support evaluation. UniMedVL simultaneously generates images and textual explanations as paired outputs, and this capability introduces new potential solutions for AI-driven assisted treatment planning tasks compared with pipelines of specialized models. Concretely, **as shown in Figure 15 in the appendix**, we provide a few visual comparisons where UniMedVL can generate images that simulate disease progression in a patient given the instructions, for example, by depicting a decrease in right perihilar infection or the development of a small right pleural effusion with mild enlargement of the cardiac silhouette. Such prognostic visualizations paired with interpretable textual explanations about potential future disease states could assist clinical experts in devising more clinically plausible treatment plans. Critically, this dual-modality approach enables UniMedVL to outperform pipelines of specialized models that lack joint counterfactual generation ability and the capacity to reveal the rationale behind decision-making processes through generated explanations.
>
> To quantitatively validate the potential clinical utility of UniMedVL, while conducting diagnostic or treatment-planning tasks with human experts is beyond the scope of the current revision timeline, we evaluate the model on CXR counterfactual generation with explanations. As shown in Table 4 below, our method achieves significantly superior performance across both dimensions compared to existing approaches with and without the explanatory text.
>
>
>
> | Method | **Counterfactual Image** | | | **Explanatory Text** | | |
> |--------|----------|----------|----------|----------|----------|----------|
> | | gFID↓ | AUROC↑ | F1↑ | BLEU-3↑ | METEOR↑ | ROUGE-L↑ |
> | BiomedJourney [1] | 36.62 | 0.8385 | 0.8411 | – | – | – |
> | CXR-IRGen | 35.39 | 0.5236 | 0.7609 | 0.0448 | 0.2115 | 0.1846 |
> | ProgEmu | 29.21 | 0.7921 | **0.8914** | 0.1241 | 0.4097 | 0.2606 |
> | **UniMedVL** | **27.17** | **0.7970** | 0.8731 | **0.2641** | **0.4486** | **0.4649** |
>
> [1] Gu Y, Yang J, Usuyama N, et al. BioMedJourney: Counterfactual Biomedical Image Generation by Instruction-Learning from Multimodal Patient Journeys. arXiv preprint arXiv:2310.10765, 2023.

---

> ### Author Response · Authors · 2025-11-22
> **Response to Question 1**
>
> ## Question 1.
>
> > Ablation on VAE Fine-tuning: The paper justifies using a frozen, general-purpose VAE with strong reconstruction metrics. However, was any experimentation done with a lightly fine-tuned medical VAE? A small ablation could clarify if there are diminishing returns or potential risks of domain-specific adaptation that were avoided.
>
> A2:  We thank the reviewer for this question regarding VAE fine-tuning. Following this suggestion, we added ablation experiments with a lightly fine-tuned medical VAE (**Table 7, Appendix A.1**). The frozen general-purpose VAE (FLUX) achieves comparable reconstruction performance to both MedITok and the fine-tuned medical VAE across all modalities, with marginal metric differences. Further fine-tuning yields no significant gains beyond the frozen FLUX VAE. We will clarify this in the revision. Designing a VAE that substantially outperforms this general-purpose baseline remains an important open problem.
>
> ***Table 7** : Reconstruction quality evaluation of pretrained VAE models on medical imaging modalities.*
>
> | Metric                         | Model                         | \(f_d\) | CFP   | CT    | CXR   | Endoscopy | HIS   | MRI   | OCT   | Ultrasound |
> |--------------------------------|-------------------------------|--------:|------:|------:|------:|----------:|------:|------:|------:|----------:|
> | **rFID (Lower is Better)**     |                               |         |       |       |       |           |       |       |       |           |
> |                                | VAE (FLUX)                    | 8       | 13.22 | 5.81  | 5.42  | 11.77     | 10.00 | 10.58 | 13.23 | 9.64      |
> |                                | Direct End-to-end VAE (FLUX)  | 8       | 14.05 | 30.59 | 23.28 | 39.56     | 44.64 | 37.95 | 17.33 | 31.58     |
> |                                | VQGAN                         | 8       | 27.22 | 15.97 | 33.57 | 27.73     | 21.33 | 67.68 | 29.48 | 18.66     |
> |                                | Emu3-VQ                       | 8       | 16.27 | 11.83 | 27.91 | 20.83     | 13.52 | 69.89 | 25.43 | 11.99     |
> |                                | MedITok                       | 16      | 14.39 | 7.88  | 22.27 | 10.66     | 6.32  | 46.54 | 17.64 | 6.55      |
> | **PSNR (Higher is Better)**    |                               |         |       |       |       |           |       |       |       |           |
> |                                | VAE (FLUX)                    | 8       | 34.58 | 37.34 | 37.09 | 35.33     | 34.50 | 34.30 | 34.58 | 33.59     |
> |                                | Direct End-to-end VAE (FLUX)  | 8       | 35.11 | 34.43 | 31.28 | 31.98     | 29.69 | 34.82 | 30.83 | 35.17     |
> |                                | VQGAN                         | 8       | 35.40 | 31.13 | 29.28 | 25.60     | 29.54 | 20.94 | 24.79 | 31.68     |
> |                                | Emu3-VQ                       | 8       | 28.96 | 36.11 | 31.68 | 28.96     | 34.32 | 22.08 | 27.57 | 35.81     |
> |                                | MedITok                       | 16      | 37.72 | 36.32 | 31.69 | 29.17     | 23.55 | 23.55 | 25.49 | 34.42     |
> | **SSIM (Higher is Better)**    |                               |         |       |       |       |           |       |       |       |           |
> |                                | VAE (FLUX)                    | 8       | 0.892 | 0.951 | 0.973 | 0.934     | 0.922 | 0.921 | 0.892 | 0.938     |
> |                                | Direct End-to-end VAE (FLUX)  | 8       | 0.842 | 0.848 | 0.904 | 0.900     | 0.938 | 0.934 | 0.867 | 0.816     |
> |                                | VQGAN                         | 8       | 0.923 | 0.885 | 0.753 | 0.768     | 0.844 | 0.484 | 0.248 | 0.317     |
> |                                | Emu3-VQ                       | 8       | 0.943 | 0.928 | 0.793 | 0.847     | 0.957 | 0.547 | 0.751 | 0.955     |
> |                                | MedITok                       | 16      | 0.953 | 0.937 | 0.855 | 0.890     | 0.972 | 0.660 | 0.935 | 0.883     |

---

### Author Response · Authors · 2025-11-26
**Global Response**

Dear Reviewers,

We sincerely appreciate the time and effort you devoted to reviewing our paper. Your detailed feedback has been invaluable, and we have made substantial revisions based on your comments to further strengthen the manuscript.

The reviewers consistently highlight several **strengths** of our work:
(i) **Conceptual novelty and unified design (Reviewers y8Tr, nhqo, hQhn, 8GAa)**: This paper proposes the clinically inspired OKA framework and present UniMedVL, a model that unifies medical multimodal understanding and generation tasks within a single, end-to-end architecture. Compared with approaches that rely on task-specific components, UniMedVL achieves superior multi-task performance through bidirectional knowledge transfer between understanding and generation and can solve diverse multimodal tasks without switching between multiple model checkpoints in clinical deployment.
(ii) **Data scale and quality (Reviewers y8Tr, nhqo, hQhn, 8GAa)**: UniMed-5M (>5.6M samples, nine modalities) as a large, diverse, and carefully curated resource with a three-stage quality-control pipeline that can substantially benefit the community.
(iii) **Training paradigm and synergy （Reviewers nhqo, 8GAa）**: The proposed Progressive Curriculum Learning and MoT backbone with shared self-attention layers facilitate bidirectional gains, where generation improves understanding and vice versa.
(iv) **Comprehensive experimental evaluation (Reviewers y8Tr, 8GAa)**: This paper conducts a comprehensive experimental evaluation across modalities and tasks. The results indicate that (a) UniMedVL achieves state-of-the-art performance on medical understanding tasks while maintaining competitive generation quality compared with medical image generation models, (b) unified training enables mutual performance enhancement between understanding and generation, and (c) the proposed framework generalizes robustly across nine imaging modalities and diverse clinical scenarios.


At the same time, we have carefully responded to the reviewers’ concerns by adding new experiments, analyses, and clarifications in our point-by-point responses to each reviewer. The major updates in the revised manuscript are as follows:

1. **Comparison with state-of-the-art medical MLLMs (Reviewer 8GAa)**.
 We now include a direct comparison with recent leading medical multimodal large language models, including **MedGemma-4B, Lingshu-7B, and Lingshu-32B**, in **Table 1 of the manuscript**. This comparison shows that UniMedVL achieves competitive or superior performance on multiple benchmarks (notably OmniMedVQA and GMAI-MMBench), while also supporting image generation and interleaved tasks that these models cannot perform.

2. **More thorough evaluation of the pretrained VAE (Reviewers y8Tr, nhqo).**
   We have significantly extended the analysis of our frozen FLUX VAE by (i) adding a comparison with a lightly fine-tuned medical VAE on our dataset across nine medical modalities(**Table 7, Appendix A.1**), and (ii) explicitly evaluating reconstruction fidelity for clinically important small lesions (**Figure 6, Appendix A.1**). These results clarify that the general-purpose VAE provides medically adequate anatomical and pathological detail and support our design choice not to heavily fine-tune a medical VAE in this work.

3. **Failure cases and qualitative analysis (Reviewer 8GAa).**
   In **Appendix A.7**, we now present representative failure cases of UniMedVL, organized by task type: medical image generation, medical image editing, and medical image understanding. For each category, we include illustrative examples and brief analyses, clarifying where the model currently struggles (e.g., subtle findings, long-horizon compositional edits) and delineating the practical boundary conditions of the model.

4. **Efficiency analysis of UniMedVL (Reviewers nhqo, hQhn).**
    In Appendix A.8, we add a dedicated efficiency analysis of UniMedVL, reporting FLOPs, throughput, latency, and peak GPU memory usage. We compare UniMedVL with BLIP3-o, Janus, and HealthGPT in Table 13 of appendix, which use a single shared multimodal transformer. The results show that, although UniMedVL utilizes a dual vision encoder and a MoT architecture, it activates only half of its parameters during inference and, in practice, still reduces FLOPs by about 3.6× for medical text-to-image generation and about 4× for VQA compared with BLIP3-o while maintaining comparable throughput.

**(Continued below....)**

---

> ### Author Response · Authors · 2025-12-03
>
> **Final Remarks:** We sincerely hope these revisions and our detailed responses adequately address the reviewers' concerns. Achieving unified comprehension and generation remains substantially more challenging due to data scarcity, domain knowledge preservation, and strict reliability requirements. Our work is, to the best of our knowledge, the first attempt to pursue unified medical multimodal learning at scale across data, model, and task levels. We construct UniMed-5M, the largest unified multimodal medical dataset to date, and design a progressive curriculum to systematically promote unified medical multimodal capabilities. Despite remaining challenges, UniMedVL achieves competitive performance on five understanding benchmarks while matching or surpassing specialised models in generation across eight medical imaging modalities. Moreover, we quantitatively illustrate bidirectional knowledge transfer. Our findings empirically support that the unified paradigm provides medical task-agnostic synergy unattainable with separate models. The approach represents a building block for future medical AI, unifying modalities like biomedical images, clinical notes, EHRs and beyond. We hope this clarifies the significance of our contribution toward scalable and unified medical multimodal AI.

---

### Note · Authors · 2026-01-27

I have read and agree with the venue's withdrawal policy on behalf of myself and my co-authors.

---

### Meta-Review · Area_Chair_5h5e · 2026-01-02

**Summary:**

This paper presents UniMedVL, a unified framework for medical multimodal understanding and generation based on the Observation-Knowledge-Analysis (OKA) paradigm. The main contributions include: (1) UniMed-5M, a large-scale dataset with 5.6M samples across nine imaging modalities; (2) Progressive Curriculum Learning with three training stages; and (3) a Mixture-of-Transformer (MoT) architecture handling both understanding and generation within a single model. The paper received scores of 6, 6, 4, and 6 from four reviewers, placing it at the borderline.

The reviewers consistently recognized several strengths: the substantial effort in constructing UniMed-5M as a valuable community resource, comprehensive experimental evaluation across multiple benchmarks, and the interesting empirical finding of bidirectional knowledge transfer between understanding and generation tasks. The authors provided extensive rebuttals with additional experiments on efficiency analysis, VAE evaluation, failure cases, and comparisons with recent models.

However, significant concerns remain regarding the limited methodological novelty and the lack of theoretical justification for the proposed framework.

**Reviewer Concerns:**

Concerns adequately addressed:

(1) Computational efficiency (Reviewer nhqo Weakness 1, Reviewer hQhn Weakness 2): The authors added Appendix A.8 with detailed efficiency analysis (Table 13), demonstrating that UniMedVL achieves approximately 3.6× fewer FLOPs for generation and 4× fewer for VQA compared to BLIP3-o, with comparable throughput. This adequately addresses concerns about the dual-encoder overhead.

(2) VAE evaluation for clinical fidelity (Reviewer nhqo Weakness 3, Reviewer y8Tr Question 1): The authors added Figure 6 showing reconstruction quality for small lesions (polyps, skin lesions, fractures) and Table 7 comparing frozen vs. fine-tuned VAE performance. These additions reasonably demonstrate that the frozen general-purpose VAE preserves clinically important details.

(3) Missing SOTA comparisons (Reviewer 8GAa Weakness 3): The authors updated Table 1 to include MedGemma-4B, Lingshu-7B, and Lingshu-32B, showing competitive performance on most benchmarks.

(4) Failure case analysis (Reviewer 8GAa Weakness 4): The authors added Appendix A.7 with representative failure cases across generation, editing, and understanding tasks, identifying boundary conditions such as text artifacts in generation and poor performance on surgical video recognition.

Concerns not adequately addressed:

Limited theoretical insights and framework-level innovation: Reviewer hQhn (Weakness 1) explicitly stated: "The paper provides little explanation of why the Observation–Knowledge–Analysis framework leads to better performance. The benefits appear mostly empirical, without any deeper analysis of internal representations or cross-modal information flow." The authors' response largely restated the three-stage pipeline with general references to curriculum learning theory, but did not provide deeper mechanistic understanding of why this particular framework works. Reviewer 8GAa confirmed this concern in their post-rebuttal comment: "although UniMedVL demonstrates strengths in data contribution and empirical results, its framework-level innovation has not fully convinced me... the level of innovation remains somewhat conservative."

Overlap with existing approaches: Reviewer hQhn (Weakness 3) noted "strong similarities to HealthGPT" and questioned "the novelty compared to HealthGPT—particularly in architecture: MoE vs MoT—is not clearly established." Reviewer 8GAa (Weakness 2) similarly observed: "Several existing works (e.g., Lingshu, HealthGPT) have already explored the benefits of multi-stage training for medical reasoning. Progressive, task-granular post-training strategies are becoming standard practice." While the authors highlighted differences in data scale and interleaved tasks, these are primarily engineering contributions rather than conceptual advances.

Performance gaps on specialized tasks: The authors acknowledged that UniMedVL underperforms on MIMIC-CXR report generation (Reviewer hQhn Question 5) and shows weak performance on fine-grained tasks like surgical video recognition and severity grading (Table 12 in Appendix). The explanation that this stems from data allocation choices actually reinforces concerns about the unified approach—spreading capacity across modalities may compromise performance where specialized expertise matters most.

Additional AC concern—Questionable premise of pan-modality unification:

Beyond the reviewers' concerns, I have reservations about the fundamental premise of unifying radically different medical imaging modalities (pathology, X-ray, fundus photography, CT, MRI, ultrasound, dermoscopy, endoscopy, OCT) into a single model. These modalities differ fundamentally in physical imaging principles, spatial resolution, diagnostic workflows, and clinical contexts. A chest radiologist and a pathologist operate in entirely different diagnostic paradigms and rarely share clinical reasoning patterns.

The paper claims "bidirectional knowledge transfer" as a key benefit, but does not provide convincing evidence or theoretical justification for why knowledge learned from one modality (e.g., chest X-rays) would meaningfully transfer to another (e.g., histopathology). The failure cases in Appendix A.7 reveal that the model struggles precisely on tasks requiring modality-specific expertise (surgical video recognition at 28.7%, severity grading at 37.2%), suggesting that the generalist approach may sacrifice clinical precision. This raises the question of whether a unified pan-modality model offers genuine advantages over well-tuned modality-specific models in realistic clinical deployment scenarios.

**Reviewer Scores:**

Reviewer y8Tr (original: 6): Would likely maintain score. Concerns about clinical decision-support evaluation and VAE ablation were addressed.

Reviewer nhqo (original: 6): Would likely maintain score at 6. The efficiency analysis and VAE evaluation addressed their technical concerns, though they may retain some reservations about the "unified" framing given the dual-track architecture.

Reviewer hQhn (original: 4): Would likely maintain score at 4 or reduce to 2. While efficiency comparisons were added, their core concerns about limited theoretical insights and overlap with HealthGPT were not substantively resolved. The authors' theoretical justification remained superficial.

Reviewer 8GAa (original: 6): The reviewer will maintain the score, as the reviewer explicitly stated in their post-rebuttal comment that he/she chose to "maintain my original positive rating". However, the reviewer noted that "although UniMedVL demonstrates strengths in data contribution and empirical results, its framework-level innovation has not fully convinced me. From the perspective of methodological contribution in a strict sense, I believe the level of innovation remains somewhat conservative."

---

### Decision · Program_Chairs · 2026-01-26

Reject